# Allosteric modulation of proton binding confers Cl⁻ activation and glutamate selectivity to vesicular glutamate transporters

Bart Borghans, Daniel Kortzak¤a, Piersilvio Longo¤b, Bettina Kolen ,
Jan-Philipp Machtens¤b, Christoph Fahlke*

Institute of Biological Information Processing, Molekular- und Zellphysiologie (IBI-1), Forschungszentrum Jülich, Jülich, Germany

¤a Current Address: Department of Pharmaceutical Sciences, University of Maryland School of Pharmacy, Baltimore, Maryland, United States of America
¤b Current Address: Institut für Neurophysiologie, Medizinische Hochschule Hannover, Hannover, Germany
* c.fahlke@fz-juelich.de

## Abstract

Vesicular glutamate transporters (VGLUTs) fill synaptic vesicles with glutamate and remove luminal Cl⁻ via an additional anion channel mode. Both of these transport functions are stimulated by luminal acidification, luminal-positive membrane potential, and luminal Cl⁻. We studied VGLUT1 transporter/channel activation using a combination of heterologous expression, cellular electrophysiology, fast solution exchange, and mathematical modeling. Cl⁻ channel gating can be described with a kinetic scheme that includes two protonation sites and distinct opening, closing, and Cl⁻-binding rates for each protonation state. Cl⁻ binding promotes channel opening by modifying the $pK_a$ values of the protonation sites and rates of pore opening and closure. VGLUT1 transports glutamate and aspartate at distinct stoichiometries: H⁺-glutamate exchange at 1:1 stoichiometry and aspartate uniport. Neurotransmitter transport with variable stoichiometry can be described with an alternating access model that assumes that transporters without substrate translocate in the doubly protonated state to the inward-facing conformation and return with the bound amino acid substrate as either singly or doubly protonated. Glutamate, but not aspartate, promotes the release of one proton from inward-facing VGLUT1, resulting in preferential H⁺-coupled glutamate exchange. Cl⁻ stimulates glutamate transport by making the glutamate-binding site accessible to cytoplasmic glutamate and by facilitating transitions to the inward-facing conformation after outward substrate release. We conclude that allosteric modification of transporter protonation by Cl⁻ is crucial for both VGLUT1 transport functions.

**Data availability statement:** All kinetic models, simulation source code, parameters, instructions, experimental datasets used in this study are available at: https://jugit.fz-juelich.de/computational-neurophysiology/vglut-kinetic-models and also mirrored at https://github.com/computational-neurophysiology/vglut-kinetic-models. The genetic fitting algorithm scripts are provided as-is to support reproducibility; however, they are not optimized for general use and may lack documentation or error handling.

**Funding:** This work was supported by the Deutsche Forschungsgemeinschaft (German Research Foundation) to Ch.F. (FA 301/15–2 as part of Research Unit FOR 2518 DynIon and FA 301/13-2 as part of the Research Unit FOR 2795 Synapses under stress). The funders had no role in study design, data collection and analysis, decision to publish, or preparation of the manuscript.

**Competing interests:** The authors have declared that no competing interests exist.

## Author summary

Fast and reliable excitatory signal transmission at glutamatergic synapses is a major determinant of the unique cognitive properties of the human brain. At such synapses, glutamate is released via exocytosis of synaptic vesicles, which contain high concentrations of glutamate. Vesicular glutamate transporters (VGLUTs) are responsible for filling synaptic vesicles; they are, however, not only highly selective low-affinity glutamate transporters, but also $Cl^-$ channels that are activated by $H^+$ and $Cl^-$ ions and voltage. In this work, we developed a computational model to describe these functions using experiments and kinetic modeling. Transporter function is based on a series of substrate association/dissociation steps and conformational transitions. To study these processes we combined current recordings in response to fast solution exchanges or voltage steps with mathematical modeling and used extensive statistical testing to identify changes in kinetic scheme parameters upon ion binding or applying voltage. Our findings provide mechanistic insights into how VGLUTs attain a different transport stoichiometry for glutamate than for aspartate and how $Cl^-$ is required for neurotransmitter transport. They also provide novel insights into the mechanisms of VGLUT anion channel opening. We expect that these results will help linking static and dynamic data about three-dimensional structures and transport functions.

## Introduction

Glutamate is the major excitatory neurotransmitter in the mammalian central nervous system. It is released from presynaptic nerve terminals via exocytosis of synaptic vesicles that have been filled by three vesicular glutamate transporters: VGLUT1, VGLUT2, or VGLUT3 [1–3]. VGLUTs employ electrochemical $H^+$ gradients generated by V-type ATPases to accumulate glutamate via stoichiometrically coupled $H^+$-glutamate exchange [4]. $H^+$-anion transport is partially uncoupled for aspartate, and the difference in coupling stoichiometry permits selective glutamate accumulation in synaptic vesicles [4]. VGLUTs also function as anion channels that mediate $Cl^-$ diffusion out of the synaptic vesicle. The distinct transport mechanisms for glutamate and $Cl^-$ permit VGLUTs to harness the outwardly directed $Cl^-$ gradient to depolarize the vesicular membrane potential and stimulate glutamate accumulation [4] and to exchange vesicular $Cl^-$ for glutamate during synaptic vesicle filling [5–7].

Glutamate transport rates are modified by the transmembrane voltage, acidic luminal pH, and chloride concentration ($[Cl^-]$) in the lumen and cytoplasm. In synaptic vesicle preparations or in liposomes containing purified and reconstituted transporters, glutamate uptake is low in the absence of $Cl^-$, but increases to maximum values at a $[Cl^-]$ of 4 mM. Higher concentrations inhibit glutamate uptake into synaptic vesicles, resulting in a biphasic $Cl^-$ dependence [8–10]. Experiments in

proteoliposomes [11] and heterologous expression systems [12] have demonstrated that—at high concentrations—Cl⁻ influx via VGLUT Cl⁻ channels affects electrogenic VGLUT glutamate transport and anion channel currents by hyperpolarizing the membrane potential. However, Cl⁻ also modulates VGLUT function as an allosteric activator, as shown in experiments using endosomal patch clamp recordings in transfected cells [13].

We combined heterologous expression of surface membrane-targeted mutant VGLUT1s and whole-cell patch clamp with fast H⁺ and Cl⁻ application and mathematical modeling to describe allosteric VGLUT activation. Our results provide first mechanistic insights into the opening as well as the voltage and ligand gating of VGLUT anion channels and into substrate selectivity and regulation of VGLUT neurotransmitter transport.

## Results

### Luminal Cl⁻ modifies the amplitude and kinetics of VGLUT1$_{PM}$ Cl⁻ currents

VGLUTs require luminal acidification for glutamate transport and anion channel activity and are activated by luminal and cytoplasmic [Cl⁻] [5]. Since only luminal [Cl⁻] undergoes major modification during synaptic function [4,7], we restricted our analysis to regulation by luminal Cl⁻. We first focused on measuring Cl⁻ currents in the absence of the transport substrate glutamate. In all experiments, we expressed a surface membrane insertion-optimized mutant of VGLUT1 (VGLUT1$_{PM}$) in HEK293T cells, followed by whole-cell patch clamping [4]. As the luminal side of surface membrane-optimized VGLUT1$_{PM}$ faces the external solution, the luminal [Cl⁻] can be varied by changing [Cl⁻] outside the cell ([Cl⁻]$_o$).

Fig 1A shows the current responses of HEK293T cells with heterologously expressed VGLUT1$_{PM}$ to voltage steps between -160 mV and +60 mV at three external [Cl⁻] (0, 40, and 140 mM) and an external pH of 5.5. In the complete absence of external Cl⁻, currents are small but clearly above background. Increased [Cl⁻]$_o$ enhances current amplitudes and accelerates the time course of activation (Fig 1A). Fig 1B depicts a plot of normalized late current amplitudes versus external [Cl⁻], which can be fitted with a Michaelis-Menten relationship providing an apparent $K_M$ of 28.3 ± 0.7 mM (mean and 95% confidence interval, with bootstrapped global fit and sampling of 1000, n = 11). Under all test conditions, VGLUT1 currents are pH dependent, with zero current amplitude at neutral external pH. Fig 1C provides the pH dependence for two different [Cl⁻]$_o$ at -160 mV, with indistinguishable p$K_M$ values of 5.3 ± 0.003 (n = 10, Hill coefficient = 1.2) at [Cl⁻]$_o$ = 0 mM and 5.4 ± 0.003 (n = 11, Hill coefficient = 1.1, p = 0.082) at [Cl⁻]$_o$ = 140 mM.

Transporters undergo conformational changes upon ligand association or dissociation, and rates for these processes can be quantified via a kinetic description of transporter currents upon fast substrate application [14–23]. Fig 1D and 1E depict normalized current responses to pH changes from 7.4 to 5.0 or 5.5 and back to pH 7.4 when the external solution is either Cl⁻ free (Fig 1D) or has a [Cl⁻]$_o$ of 140 mM (Fig 1E). Fast external solution exchange was achieved by the piezo-driven movement of a perfusion pipette made from dual-channel theta glass tubing [24–27]. Non-transfected HEK293T exhibit only negligible background currents [4] (S1 Fig).

External Cl⁻ accelerates the time course of both processes. Whereas VGLUT1$_{PM}$ Cl⁻ currents activate on monoexponential time courses and deactivate on biexponential time courses upon acidification with a Cl⁻-free external solution, the activation is biexponential and deactivation is monoexponential when the [Cl⁻]$_o$ = 140 mM. The application of external Cl⁻ results in biexponential increases and its removal in monoexponential current deactivation (Fig 1F). Time constants were only minimally voltage dependent and were comparable with and without external Cl⁻. Activation and deactivation time constants were within the same order of magnitude and largely voltage independent (S2 Fig). For Cl⁻ channels, rapid changes in external [Cl⁻] will affect the driving force and thus the unitary current amplitude. However, since our recordings are performed at voltages far away from the current reversal potential, fast, diffusion-limited changes were not observed. Current responses to fast Cl⁻ application are slow (Fig 1F), since they are based on conformational changes that modify the open probability of the channel.

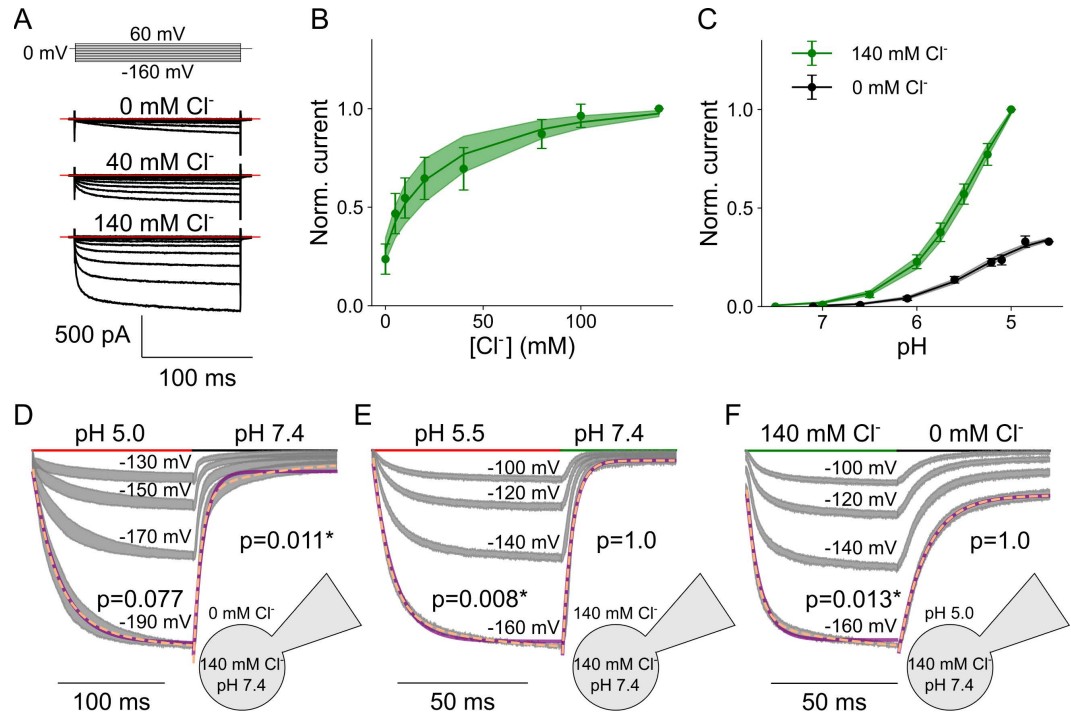

**Fig 1. WT VGLUT1$_{PM}$ chloride currents are modulated by voltage, pH, and [Cl⁻].** **(A)** Representative WT VGLUT1$_{PM}$ Cl⁻ current responses to voltage steps between -160 mV and +60 mV at pH 5.0 and external [Cl⁻] of 0, 40, and 140 mM. **(B)** Dose-response plots for VGLUT1$_{PM}$ Cl⁻ currents at pH 5.0 and rising [Cl⁻]$_o$ (means ± confidence interval, n = 11 cells, fitted with Michaelis-Menten relationships for a $K_M$ of 28.3 ± 0.7). **(C)** Dose-response plots for VGLUT1$_{PM}$ Cl⁻ currents with rising pH at either [Cl⁻]$_o$ = 0 (black, means ± confidence interval, n = 10 cells, fitted with Hill relationships for a p$K_M$ of 5.3 ± 0.003 and a Hill coefficient of 1.2) or 140 mM (green, means ± confidence interval, n = 11 cells, fitted with Hill relationships for a p$K_M$ of 5.4 ± 0.003 and a Hill coefficient of 1.1; lines and shaded areas depict the mean and 95% confidence interval). **(D-F)** Normalized chloride current responses to pH jumps from pH 7.4 to 5.0 at [Cl⁻]$_o$ = 0 mM **(D)**, from pH 7.4 to 5.5 at [Cl⁻]$_o$ = 140 mM **(E)**, or [Cl⁻] jumps from 0 mM to 140 mM and back at pH 5.5 **(F)**. Currents were recorded at four continuous voltage levels and shown as 95% confidence interval (gray) from at least 11 experiments, with currents at the most negative voltage fitted with a single (purple line) and double (dashed orange line) exponential functions provided with F-test p-values for each solution change to indicate whether they are better described by biexponential fits, with asterisks marking those that are.

## A kinetic scheme to describe VGLUT1 anion channel function

VGLUT1$_{PM}$ Cl⁻ currents activate and deactivate upon pH jumps between 5.0 and 7.4, with comparable time constants (S2 Fig). This result excludes the possibility of direct channel activation by H⁺ and indicates that protonated and unprotonated transporters can both open and close, but at distinct rates [28]. A four-state model (with one open and one closed state for each protonation state) predicts monoexponential activation and deactivation upon pH steps. Therefore, we added one doubly protonated open and one doubly protonated closed state to account for biexponential activation or deactivation in response to pH steps. To describe faster activation and deactivation in the presence of external Cl⁻, we assumed that channels with bound Cl⁻ activate or deactivate at different rates and added Cl⁻-bound open and closed states for each protonated state, resulting in a 12-state model (Fig 2). Cl⁻ binding is expected to be diffusion limited, and we combined Cl⁻ binding and associated processes into single rate constants.

This kinetic scheme accurately predicts VGLUT1$_{PM}$ anion channel gating in the presence or absence of external Cl⁻. Fig 3 depicts the simultaneous fits to normalized current responses from at least 10 HEK293T cells expressing wild-type (WT) VGLUT1$_{PM}$ to voltage steps either from a holding potential of 0 mV (Fig 3A and 3D) or after a prepulse to a more negative voltage (Fig 3B and 3E), or to pH steps using piezo-driven pipette movements (Fig 3C and 3F).

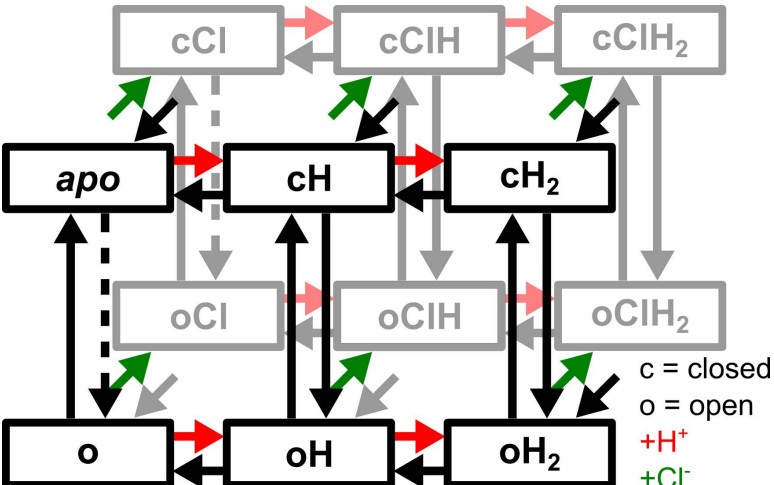

**Fig 2. Kinetic model to describe gating of VGLUT1 Cl⁻ channels.** Green arrows depict Cl⁻ binding to the black states in front, red arrows depict protonation, and ligand unbinding is shown in corresponding black arrows. The six states on top are *apo* and other closed states, and the channel opens via black downward arrows. Dashed arrows show rate-limited opening without protonation.

Experiments were performed in the absence of external Cl⁻ (Fig 3A–C) or at $[Cl^-]_o = 140$ mM (Fig 3D–F); Fig 3G shows the fits to current responses upon changes in $[Cl^-]_o$ at an external pH of 5.5, and Fig 3H and 3I depict the $[Cl^-]$ and the pH dependence of normalized late currents. To ensure global model optimization and to assess how well the individual parameters are defined by our experimental data, we randomly modified the fit parameters using an explorative genetic algorithm and collected parameter set values that impaired the goodness of fit by < 25% for each simulated experimental time course and metric (we denote parameter-specific simulation output that results in a calculated value, such as open probability, open time, transport rate, $pK_a$ values, pH and $[Cl^-]$ dependence as metrics). From these sets of parameters, model predictions from 250 fits are shown as blue lines, evenly spaced from ≥ 10,000 individuals with ≥ 3000 unique values for each variable to provide reliable statistical distinction. Kinetic modeling enables the calculation and visual tracking of overall fluxes through the states of the kinetic model and, thus, provides insight into activation and deactivation pathways. Fig 3J shows the kinetic model with the occupancy of individual states depicted as the circle size and rate amplitude as the thickness of curved arrows in the absence or presence of external Cl⁻, and Fig 3K residence probabilities for the transporter states under both conditions. In the absence of Cl⁻, VGLUT1_PM predominantly opens from the singly protonated closed state (as shown in Fig 3J) and remains singly protonated in the open state (Fig 3K).

Fig 3L depicts the three most frequent activation and deactivation pathways at pH 5.5, 140 mM Cl⁻, and -160 mV, according to transition path theory-based reactive flux analysis. Under these conditions, VGLUT1_PM anion channels predominantly activate via initial Cl⁻ binding, followed by protonation and subsequent channel opening into a singly protonated conformation (oClH). Low-affinity Cl⁻ binding and rapid unbinding make the Cl⁻-free singly protonated state (oH) the most common open state (Fig 3K). There is a much lower likelihood of channels opening after additional protonation from the doubly protonated Cl⁻ bound open state (oClH₂). Kinetic modeling also assigns a small probability to WT VGLUT1 opening without protonation (*apo*-cCl-oCl); however, this open state is unstable, leading to negligible channel opening at neutral pH, in agreement with current at pH 7.4 being indistinguishable from background [4]. Channel deactivation proceeds primarily via three pathways with comparable probabilities: by closing in the singly protonated state, followed by deprotonation; after deprotonation in the open state; or from a doubly protonated Cl⁻-bound state (Fig 3L).

Fig 4A depicts the $pK_a$s for each of the two protonation sites for both Cl⁻-free and Cl⁻-bound VGLUT1 at 0 mV, and at -160 mV; the parameter sets that collectively describe the experimental data in an adequate fit, in the following denoted as

PLOS Computational Biology

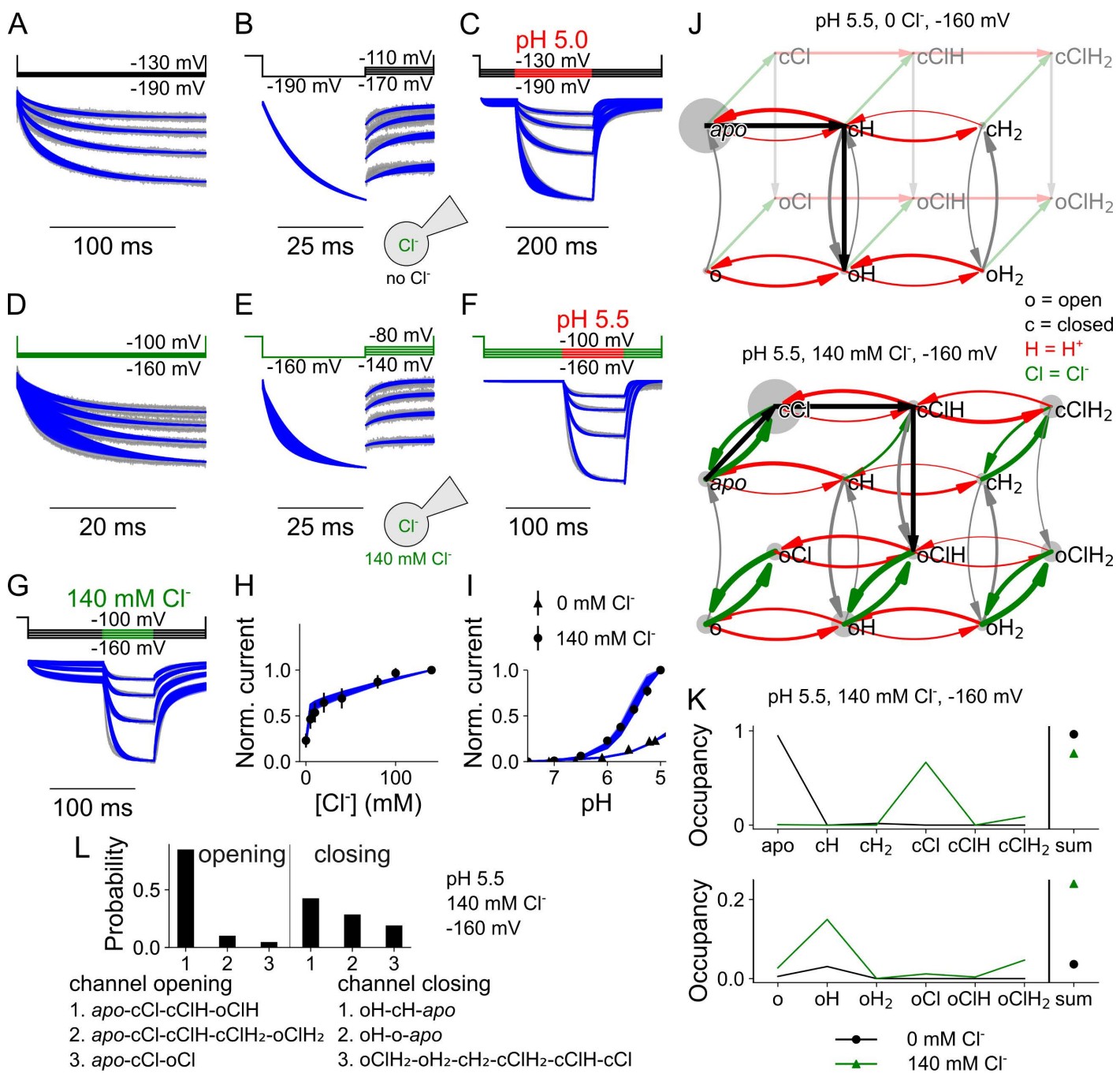

**Fig 3. VGLUT1 Cl⁻ currents are well described by a kinetic scheme that assumes three protonation states, each with a closed and open anion channel conformation and all of which can bind Cl⁻.** (A–I) Current responses and predictions from the kinetic scheme for voltage steps from 0 mV to negative potentials with $[Cl^-]_o = 0$ **(A)**, voltage steps from -190 mV to less negative potentials with $[Cl^-]_o = 0$ **(B)**, rapid pH steps to pH 5.5 at $[Cl^-]_o = 0$ **(C)**, voltage steps from 0 mV to negative potentials with $[Cl^-]_o = 140$ mM **(D)**, voltage steps from -160 mV to less negative potentials with $[Cl^-]_o = 140$ mM **(E)**, rapid pH steps to pH 5.5 at $[Cl^-]_o = 140$ mM **(F)**, rapid Cl⁻ steps to 140 mM at pH 5.0 **(G)**, late current plots versus $[Cl^-]_o$ (means ± 95% confidence interval; **H**), late current plots versus pH at $[Cl^-]_o = 0$ and 140 mM **(I)**. **(J)** Kinetic schemes describing VGLUT1 Cl⁻ channel activation, with circle size giving occupancy of individual states and curved arrow thickness the rate amplitude for $[Cl^-]_o = 0$ (top) and $[Cl^-]_o = 140$ mM (bottom) at pH 5.5 and -160 mV. **(K)** Simulated residence probabilities for the indicated closed (top) and open (bottom) states for VGLUT1 anion channels with (green) or without (black) bound Cl⁻. **(L)** Three most frequently occurring pathways for activation from *apo* (left) or deactivation from the most common two open states (oH and oClH₂) to an unprotonated closed state (*apo* or cCl, right). Experimental data are given as the 95% confidence interval of patch clamp data (gray) and the fitting results of 250 simulations generated through exploratory mutations (blue) under the same conditions.

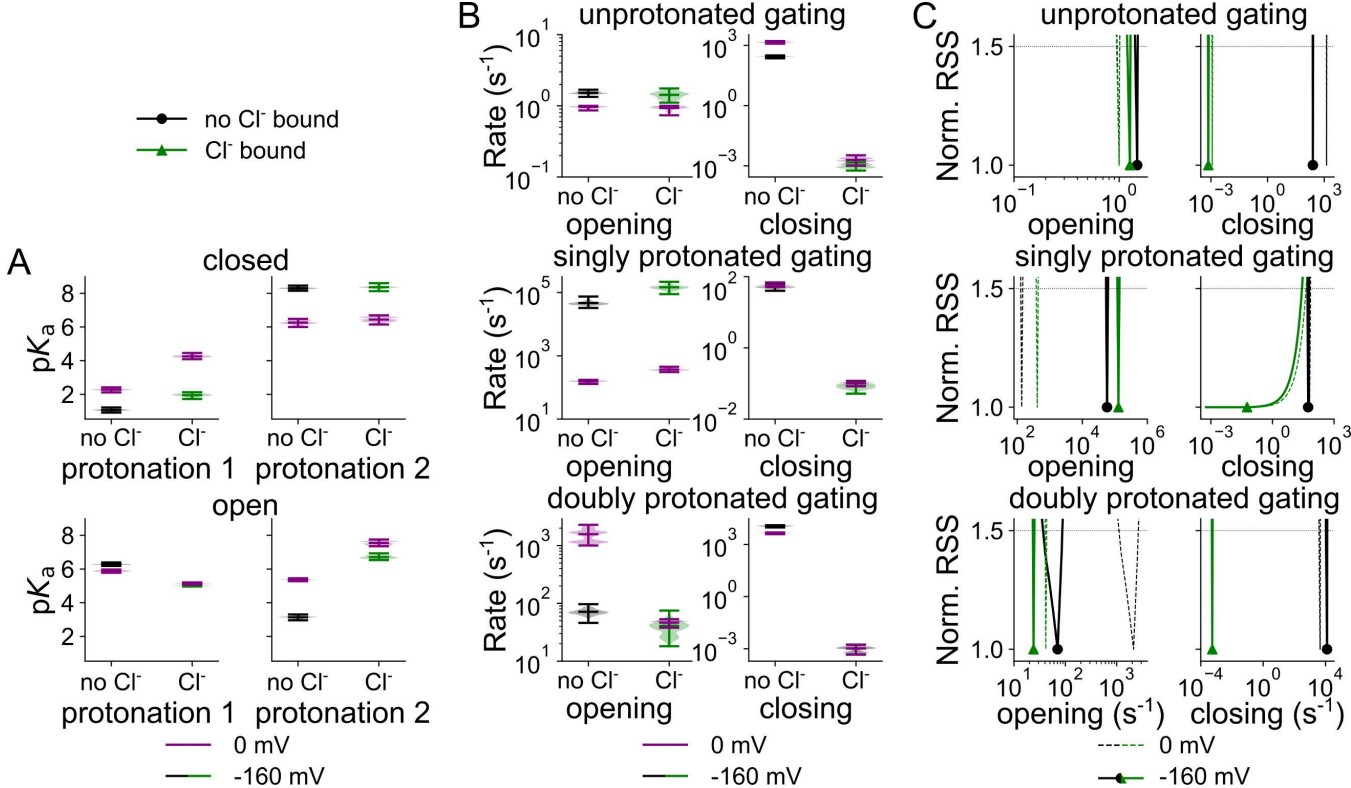

**Fig 4. Cl⁻ binding changes VGLUT1 protonation and opening/closing rates. (A)** Cl⁻-dependent changes in predicted p$K_a$s of the first and second protonation sites of the closed (top) or open (bottom) channel at -160 mV (green, Cl⁻ bound; black, no Cl⁻) or at 0 mV (purple or dashed lines). **(B)** Violin plots of the amplitude range generated by exploratory mutation. **(C)** Rates for channel opening and closing, depicted as normalized RSS representing goodness of fit for a range of amplitudes.

parameter variants, are shown as violin plot. The z and d parameters that describe the voltage dependence of all values in Fig 4 are given in S3 Fig, and an overview of all optimized parameters for WT VGLUT1$_{PM}$ is provided in S1 Table. Negative voltages reduce the *apo* state p$K_a$ from a median of 2.3 (for a total amplitude range of 2.1–2.4) to 1.1 (0.9–1.2) and after Cl⁻ binding from 4.3 (4.1–4.5) to 2.0 (1.7–2.1). For the second proton in the closed state, the voltage increases the p$K_a$ by roughly 2 points, regardless of Cl⁻. This rise, from 6.3 (6.0–6.5) to 8.3 (8.2–8.5) without and from 6.4 (6.1–6.7) to 8.4 (8.1–8.6) with Cl⁻ bound, combined with opening from a single protonated state, explains the relatively high occupancy of cClH$_2$ while VGLUT1 is closed (Fig 3K).

Titratable amino acids exhibit p$K_a$ values above 2.0 in solution. Although p$K_a$ values can shift in proteins, such changes always result in higher and not in lower values. Kinetic modeling predicted several p$K_a$ below 2.0, and these low values are likely due to minimization of the state number, which makes lumping several separate reactions into one state necessary. Apparent p$K_a$s of such compound protonation reaction can deviate from the p$K_a$ describing protonation of the amino acid side because of additional energy-consuming steps that changes the accessibility of the titratable side chain.

Fig 4B and 4C show opening and closing rates with and without external Cl⁻ at 0 mV or -160 mV. In addition to comparing the full range of parameters obtained in the exploratory fitting (Fig 4B), we also applied a statistical test to calculate the change in the quality of fit upon only varying the compared parameter (Fig 4C; [27]). In such a plot, a sharply defined and distinct parameter value is associated with instantly increasing RSS upon small changes in parameter values without overlap with parameter values obtained under other conditions. The two tests—RSS goodness of fit by parameter

amplitude and amplitude distribution via exploratory mutation—demonstrate that—except for closure of the singly protonated channel—all opening and closing transitions are well defined by the fitting procedure and undergo a statistically significant change upon the application of Cl⁻. At -160 mV, Cl⁻ increases opening rates from $4.6 \times 10^4$ ($3.3 \times 10^4$–$7.5 \times 10^4$, median and parameter amplitude range) s⁻¹ to $1.5 \times 10^5$ ($9.0 \times 10^4$–$2.2 \times 10^5$) s⁻¹ in the singly protonated state, but reduces it from 71 (46–97) s⁻¹ to 41 (18–75) s⁻¹ in the doubly protonated state. The membrane voltage seems to dictate the preference for opening singly protonated, with otherwise low opening rates of 157 (129–174) s⁻¹ and 361 (305–451) s⁻¹ without and with Cl⁻, respectively. Membrane hyperpolarization decreases the opening rates for doubly protonated VGLUT1 in the absence, but not in the presence of Cl⁻ (Fig 4B and 4C). Cl⁻ reduces the closing rates for all protonation states: at -160 mV from 277 (228–319) s⁻¹ to $9.9 \times 10^{-4}$ ($5.8 \times 10^{-4}$–$1.5 \times 10^{-3}$) s⁻¹ without protonation, from 51 (40–64) s⁻¹ to 0.08 (0.05–0.11) s⁻¹ for the singly protonated state, and from $1.1 \times 10^4$ ($9.4 \times 10^3$–$1.3 \times 10^4$) s⁻¹ to $1.1 \times 10^{-3}$ ($4.7 \times 10^{-4}$–$1.8 \times 10^{-3}$) s⁻¹ for the doubly protonated state. The membrane voltage has only minor effects on channel closure.

Cl⁻ binding and unbinding depend on the protonation state and differ for the open and closed conformations (Fig 5, corresponding $z$ and $d$ parameters are given in S4 Fig). Cl⁻ association with open channels is close to the limits of diffusion control ($10^9$ M⁻¹ s⁻¹; [29]. Whereas closed unprotonated and doubly protonated channels bind Cl⁻ at very high rates, at -160 mV with concentration-normalized rates of $6.7 \times 10^8$ ($5.8 \times 10^8$–$7.2 \times 10^8$) M⁻¹ s⁻¹ and $1.1 \times 10^7$ ($9.4 \times 10^6$–$1.3 \times 10^7$) M⁻¹ s⁻¹, respectively, we observed much lower association rates for the singly protonated closed channel of $1.8 \times 10^3$ ($1.6 \times 10^3$–$2.1 \times 10^3$) M⁻¹ s⁻¹. Such association rates may indicate that Cl⁻ binding requires conformational changes that make the binding site accessible in this protonation state.

Since changes in the Cl⁻-binding rate will also affect the likelihood of binding-site occupation, it was necessary to distinguish whether protonation adjusts transporter function via altering the binding affinity or association rate for Cl⁻. Therefore, we also assessed how modification of both the association and dissociation rates of Cl⁻ (when held at a fixed ratio to maintain the Cl⁻ dissociation constant, $K_D$) affect the goodness of fit (Fig 5B). The analysis demonstrated that only the rate of Cl⁻ association with the singly protonated closed channels determines the fit quality, while for other states such differences could not be demonstrated (Fig 5). Moreover, with constant $K_D$, only singly protonated Cl⁻ in the closed state is sharply defined, showing the individual association rates are more important here than for any of the other states. We conclude

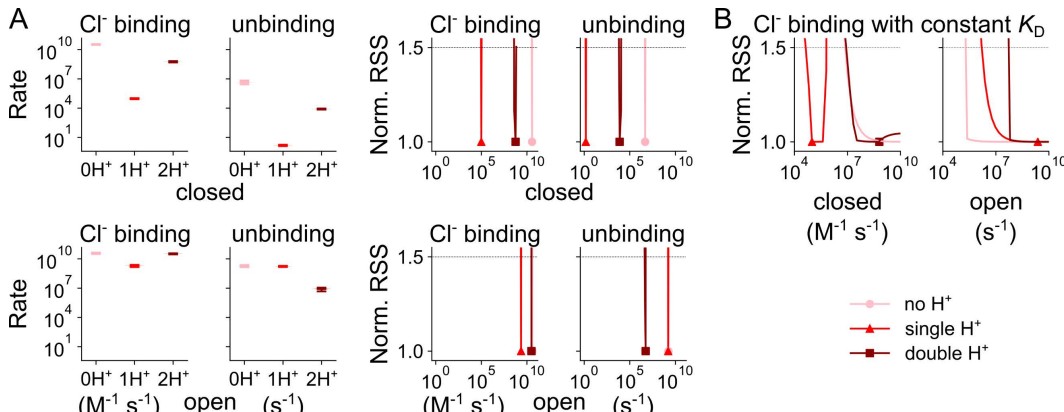

**Fig 5. Changes in Cl⁻ association with WT VGLUT1_PM with protonation state. (A)** Secondary Cl⁻-binding rates to closed (top) or open (bottom) WT VGLUT1_PM anion channels in the unprotonated state or the singly or doubly protonated state (light→dark red indicates increasing protonation). Rates are depicted as violin plots (left) showing the amplitude range generated by exploratory mutation and normalized RSS (right) representing goodness of fit for a range of amplitudes. **(B)** Cl⁻ binding rates at constant $K_D$, where the unbinding constants were simultaneously altered to maintain the Cl⁻-binding affinity. All rates are shown at -160 mV.

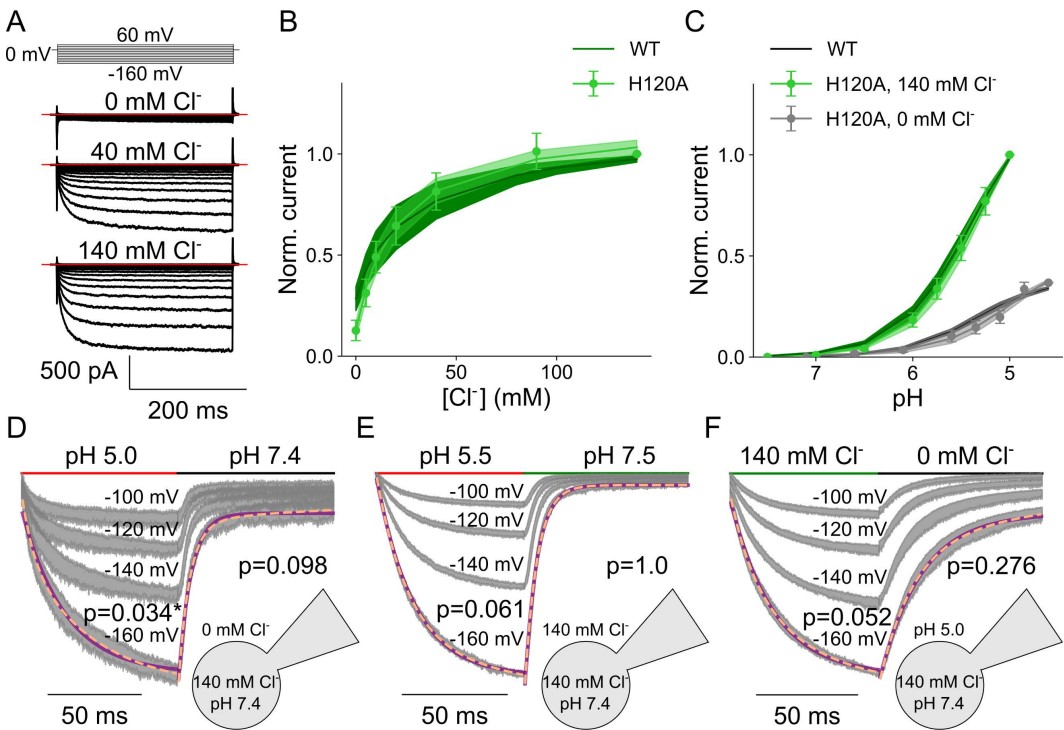

**Fig 6. H120A VGLUT1$_{PM}$ chloride currents are modulated by voltage, external pH, and [Cl$^-$].** **(A)** Representative H120A VGLUT1$_{PM}$ Cl$^-$ current responses to voltage steps between -160 mV and +60 mV at pH 5.0 and external [Cl$^-$] of 0, 40, or 140 mM. **(B)** Dose-response plots for VGLUT1$_{PM}$ Cl$^-$ currents at rising [Cl$^-$]$_o$ at pH 5.0 (means ± confidence interval, n = 11 cells, fitted with Michaelis-Menten relationships for a $K_M$ of 19.4 ± 0.2. **(C)** Dose-response plots for VGLUT1$_{PM}$ Cl$^-$ currents for rising pH at [Cl$^-$]$_o$ = 0 (black, means ± confidence interval, n = 11 cells, fitted with Hill relationships for a $pK_M$ of 5.0 ± 0.004 and a Hill coefficient of 1.2 for H120A) or 140 mM (green, means ± confidence interval, n = 11 cells, lines and shaded areas depict the mean and 95% confidence interval, fitted with Hill relationships for a $pK_M$ of 5.4 ± 0.003 and a Hill coefficient of 1.3 for H120A). **(D-F)** Normalized chloride current responses to pH jumps from pH 7.4 to 5.0 at [Cl$^-$]$_o$ = 0 mM **(D)**, or from pH 7.4 to 5.5 at [Cl$^-$]$_o$ = 140 mM **(E)**, or [Cl$^-$] jumps from 0 mM to 140 mM and back at pH 5.5 **(F)** held at four continuous voltage levels. Solution change time courses are shown as the 95% confidence interval (gray) from at least 11 experiments; those at the most negative voltage are fitted with a single (purple line) and double (dashed orange line) exponential function and are provided with F-test p-values to indicate whether they are better described by biexponential fits, with asterisks marking those that are.

that during anion channel activation VGLUT1 transiently assumes conformational states that are inaccessible to anions in the external solution. Such conformations might be inward facing or occluded with the Cl$^-$-binding site inaccessible from either membrane side.

## H120A impairs Cl$^-$ binding to VGLUT1

Substitution of histidine at position 120 by alanine changes the permeation and gating of VGLUT1$_{PM}$ anion channels (Fig 6A; [4]). We found that the H120A mutation modifies the Cl$^-$ dependence of late current amplitudes (H120A: $K_M$ = 19.4 ± 0.2 mM, WT: $K_M$ = 28.3 ± 0.7 mM, mean and 95% confidence interval, p = 0.047; Fig 6B) and changes the $pK_M$ of acid activation from 5.3 ± 0.003 (WT, Hill coefficient = 1.2) to 5.0 ± 0.004 (H120A, Hill coefficient = 1.2, p = 2.2 × 10$^{-6}$) in the absence of Cl$^-$, but not in the presence of Cl$^-$ (5.4 ± 0.003 and Hill coefficient = 1.3, compared to 5.4 ± 0.003 with a Hill coefficient = 1.1 for the WT; Fig 6C).

H120A decelerates VGLUT1$_{PM}$ kinetics in voltage steps (Fig 6A), pH steps (Fig 6D and 6E), and Cl$^-$ steps (Fig 6F). Unlike WT, H120A VGLUT1$_{PM}$ Cl$^-$ currents activate and deactivate over a monoexponential time course upon acidification with external Cl$^-$. In the absence of external Cl$^-$, the kinetics of WT and H120A VGLUT1$_{PM}$ current activation/deactivation

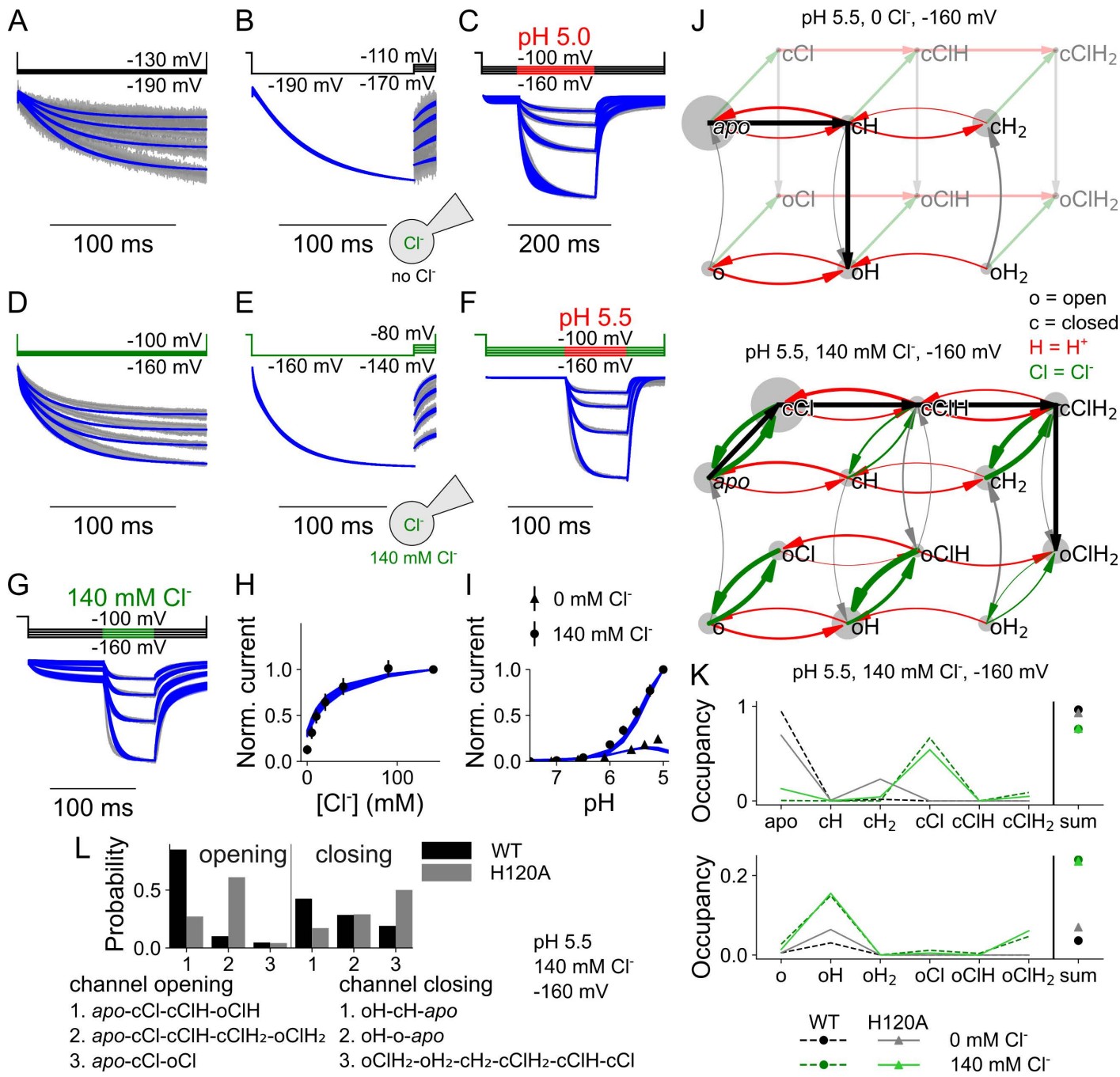

**Fig 7. H120A VGLUT1 Cl⁻ currents are well described by a kinetic scheme that assumes three protonation states, each with a closed and open anion channel conformation and all of which can bind Cl⁻.** (A–I) Current responses and predictions by the kinetic scheme for voltage steps from 0 mV to negative potentials at $[Cl^-]_o = 0\,mM$ **(A)**, more negative to less negative potentials at $[Cl^-]_o = 0$ **(B)**, pH steps from pH 7.4 to pH 5.5 at $[Cl^-]_o = 0$ **(C)**, voltage steps from 0 mV to negative potentials at $[Cl^-]_o = 140\,mM$ **(D)**, more negative to less negative potentials at $[Cl^-]_o = 140\,mM$ **(E)**, pH steps from pH 7.4 to pH 5.5 at $[Cl^-]_o = 140\,mM$ **(F)** or upon changes in $[Cl^-]_o$ from 0 to 140 mM at pH 5 **(G)**, plots of late currents (means ± 95% confidence interval) versus $[Cl^-]_o$ **(H)** or pH **(I)**. **(J)** Kinetic schemes describing H120A VGLUT1 Cl⁻ channel activation for $[Cl^-]_o = 0\,mM$ (top) and $[Cl^-]_o = 140\,mM$ (bottom, with circle size giving occupancy of individual states and curved arrow thickness the rate amplitude. **(K)** Simulated residence probabilities for the indicated closed (top) and open (bottom) states for H120A VGLUT1 anion channels with (green) or without (black) bound Cl⁻. **(L)** Three most frequently occurring activation (left) or deactivation (right) pathways upon a rapid pH change. Experimental data are given as the 95% confidence interval of patch clamp data (gray) and the fitting results of 250 simulations from parameter sets generated through exploratory mutations (blue) under the same conditions.

upon pH steps are comparable; at high external [Cl⁻], H120A decelerates activation and deactivation (S5 Fig). Changes in the external [Cl⁻] resulted in monoexponential increases and decreases in the H120A VGLUT1 current (Fig 6F) at time courses much slower than those of the WT (S5 Fig); for all conditions, time constants were only minimally voltage dependent.

We reevaluated the parameter amplitudes with the 12 state model developed for WT VGLUT1$_{PM}$, until H120A VGLUT1$_{PM}$ Cl⁻ currents were described well (Fig 7A–I). Fig 7J depicts the overall flux through the states of the kinetic model, and Fig 7K the occupancy of individual states. In the absence of Cl⁻, H120A VGLUT1$_{PM}$ anion channels predominantly open in the singly protonated state, with higher probabilities than the WT; in the presence of Cl⁻, occupation of oClH$_2$ was slightly higher for H120A than for WT (Fig 7K). The H120A mutation alters the activation and deactivation pathways of VGLUT1 channels (Fig 7L, paths numbered by WT preference). Mutant anion channels primarily open from the doubly protonated state (cClH$_2$, path 2), with a lower probability of opening from the singly protonated Cl⁻-bound state. Whereas the WT mainly closes from the oH state, either by direct deprotonation or by deprotonation of the closed state, H120A VGLUT1$_{PM}$ mainly passes through Cl⁻ unbinding to oH$_2$ and then to cH$_2$, cClH$_2$, cClH, and, finally, to cCl (path 3).

An overview of all optimized parameters for H120A VGLUT1$_{PM}$ is provided in S2 Table. Without Cl⁻, H120A strongly increases the p$K_a$ of the first protonation site of the closed channel from 1.1 (0.9–1.2), median and parameter amplitude range) to 3.7 (3.5–3.9; all values obtained at -160 mV; Fig 8A). Cl⁻ binding reduces this value back to 3.1 (2.8–3.2). H120A impairs protonation of the second site, without effect of Cl⁻. For Cl⁻-bound open VGLUT1, H120A decreases the p$K_a$ of the first protonation site from 5.0 (5.0–5.1) to 2.7 (2.6–3.0), with a smaller increase from 6.7 (6.5–6.9) to 7.8 (7.6–8.0) for the second protonation site. H120A decreases opening rates for singly protonated states, from $4.6 \times 10^4$ ($3.3 \times 10^4$–$7.5 \times 10^4$) s⁻¹ to 74 (55–93) s⁻¹ without Cl⁻ and from $1.5 \times 10^5$ ($9.0 \times 10^4$–$2.2 \times 10^5$) s⁻¹ to 1439 (1044–1807) s⁻¹ with Cl⁻ (Fig 8B). It increases closing rates in the presence of Cl⁻: from $9.9 \times 10^{-4}$ ($5.8 \times 10^{-4}$–$1.5 \times 10^{-3}$) s⁻¹ to 0.026 (0.012–0.050) s⁻¹ when unprotonated, from $1.1 \times 10^{-3}$ ($4.7 \times 10^{-4}$–$1.8 \times 10^{-3}$) s⁻¹ to 1.0 (0.5–1.5) s⁻¹ when doubly protonated; with single protonation, the rates are not sharply defined. Moreover, for doubly protonated states, H120A has a lower opening rate than the WT in the absence of Cl⁻ with 4.2 (2.7–5.7) s⁻¹ versus 71 (46–97) s⁻¹, but a higher rate in its presence with 250 (85–532) s⁻¹ versus 41 (18–75) s⁻¹. This likely forms the basis for its preferential channel opening under double protonation only in the presence of Cl⁻. H120A alters the preferred pathways of channel activation through changes in channel opening and closing, effectively switching its primary Cl⁻-activated opening state from single to double protonation. S6 Fig depicts the effects of the H120A mutation on the voltage dependence of protonation, opening/closing, and Cl⁻ association in the absence and presence of luminal/external Cl⁻.

The H120A mutation has pronounced effects on Cl⁻ binding and unbinding (Fig 8C). Whereas H120A left Cl⁻ dissociation constants for the unprotonated open channel unaffected, it increased the $K_D$ under all other conditions, in the open channel from 869 (641–1263) mM to $5.8 \times 10^5$ ($2.9 \times 10^5$–$8.3 \times 10^5$) mM when singly protonated, and from 0.24 (0.15–0.36) mM to 3.1 (2.0–4.5) mM when doubly protonated. For the closed channel, dissociation constants increase from 0.12 (0.08–0.20) mM to 3.4 (2.2–5.8) mM when unprotonated, from 0.017 (0.013–0.019) mM to 16 [11–22] mM when singly protonated, and from 0.014 (0.012–0.017 mM to 38 (11–85) mM when doubly protonated. These changes correspond to altered Cl⁻ association rates (Fig 8C). However, by testing the variation in Cl⁻ binding to open or closed channel at a fixed affinity (S7 Fig), we demonstrate that H120A does not affect the accessibility but, rather, the affinity of the binding sites.

## A kinetic scheme to describe VGLUT1 glutamate and aspartate transport

VGLUT1$_{PM}$ glutamate or aspartate currents are much smaller than Cl⁻ currents, requiring complete substitution of intracellular Cl⁻ with these anions to record such transport currents [4]. Fig 9 depicts representative glutamate and aspartate currents, as well as the pH and [Cl⁻] dependence of late currents at -160 mV, with S8 Fig showing the time constants of pH and [Cl⁻] jumps. Aspartate currents differ from glutamate, with a slightly higher amplitude and a more negative current reversal potential (Fig 9A and 9B), reflecting the difference in transport stoichiometry [4]. Glutamate and aspartate

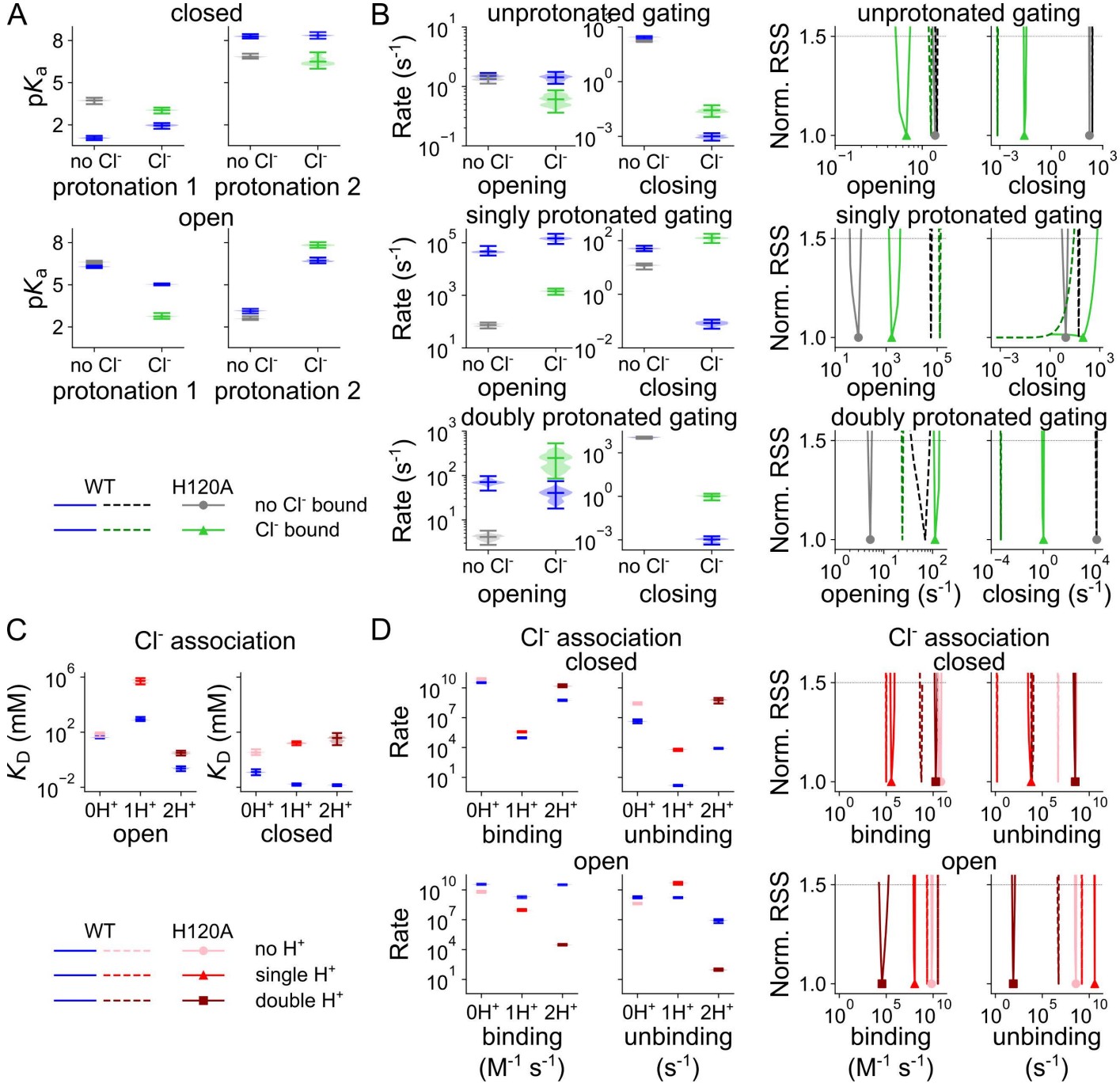

**Fig 8. H120A VGLUT1_PM affects protonation and opening/closing rates as well as Cl⁻ binding.** (A) Cl⁻-dependent changes in p$K_a$ for the first and second protonation sites of the closed (top) or open (bottom) H120A VGLUT1_PM channels. (B) Opening and closing rates (no Cl⁻ in black, Cl⁻-bound in green, WT in blue or dashed lines) for different protonation states. (C) Cl⁻ association $K_D$ at different protonation states (light→dark red indicates increasing protonation). (D) Cl⁻ binding and unbinding rates for closed and open H120A VGLUT1_PM channels at different protonation states. Rates are depicted as violin plots (left) showing the amplitude range generated by exploratory mutation and normalized RSS (right) representing goodness of fit for a range of amplitudes. All rates are shown at -160 mV.

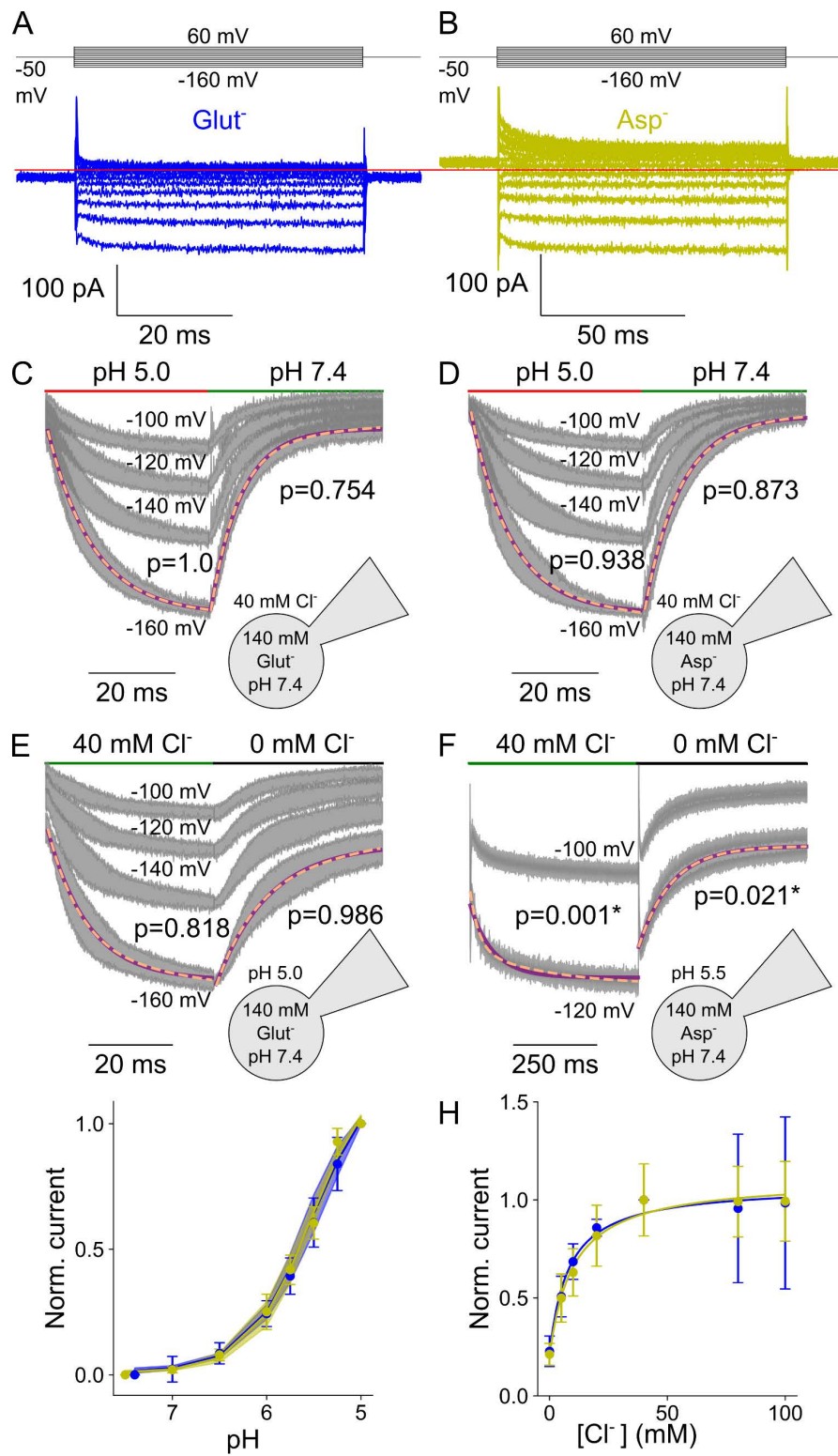

**Fig 9. VGLUT1$_{PM}$ glutamate and aspartate current are modulated by voltage, external pH, and [Cl⁻].** (A, **B**) Representative current responses of transfected cells dialyzed with glutamate-based (blue; A) or aspartate-based (yellow; B) solutions to voltage steps with [Cl⁻]$_o$ of 40 mM and the pH of 5.5, **(C - f)** Current responses to pH steps from pH 7.4 to pH 5.0 with glutamate (C) or aspartate **(D)**, or to concentration steps from [Cl⁻]$_o$ = 0 mM

to 40 mM with glutamate (E) or aspartate **(F)**. (G) pH dependence of normalized steady-state current amplitudes at $[Cl^-]_o = 140$ mM and a membrane potential of -160 mV bootstrapped using a Hill relationship with glutamate ($pK_M = 5.5 \pm 0.005$, Hill coefficient = 1.3) and aspartate ($pK_M = 5.5 \pm 0.002$, Hill coefficient = 1.4, $[Cl^-]_o = 40$ mM), **(H)** $[Cl^-]$ dependence of steady-state current means at pH 5.0 and -160 mV fitted to a Michaelis-Menten relationship with glutamate ($K_M = 8.1 \pm 2.0$ mM) and aspartate ($K_M = 9.9 \pm 2.2$ mM; pH 5.5, -140 mV). Currents (gray) and error bars represent the 95% confidence interval from ≥ 10 measurements; currents at the most negative voltage are fitted with a single (purple line) and double (dashed orange line) exponential function along with F-test p-values to indicate whether they are better described by biexponential fits, with asterisks marking those that are..

currents are close to background in the absence of $Cl^-$; therefore, we only recorded current responses to pH steps at $[Cl^-]_o = 40$ mM (Fig 9C and 9D) or to $[Cl^-]$ steps at acidic pH (Fig 9E and 9F). For glutamate and aspartate, pH jumps resulted in monoexponential activation or deactivation, with closely similar kinetics (Fig 9C and 9D). The pH dependence of late glutamate and aspartate currents can be fitted with the Hill equation, and the $pK_M$ values for both are indistinguishable (glutamate: $5.5 \pm 0.005$ with a Hill coefficient of 1.3, aspartate: $5.5 \pm 0.002$ with a Hill coefficient of 1.4, p = 0.53, mean and 95% confidence interval; Fig 9G) and similar to the values obtained with $Cl^-$ currents.

Cl$^-$ concentration jumps elicited very different responses for the glutamate and aspartate currents. Rapid increases in $[Cl^-]_o$ from 0 to 40 mM resulted in monoexponential increases of glutamate currents with time constants of around 10 ms, and the currents decayed upon decreasing steps to $[Cl^-]$ of 0 mM over a similar time course (Fig 9). For aspartate currents, Cl$^-$-dependent activation was more than 10-fold slower, and followed a biexponential time course upon $[Cl^-]$ increases or decreases with time constants of around 140 ms (Figs 9F and S8B). Despite these differences in kinetics, glutamate and aspartate currents rose with increasing external $[Cl^-]$ with a similar concentration dependence (i.e., glutamate: $K_M = 8.1 \pm 2.0$ mM, aspartate: $9.9 \pm 2.2$ mM, mean and 95% confidence interval; Fig 9H). These values are significantly lower than the $K_M$ of $28.3 \pm 0.7$ mM for VGLUT1 Cl$^-$ currents (p = 0.0097, n ≥ 10 cells, one-sample Wilcoxon signed rank test between the combined glutamate mean fit and individual cellular fits of Cl$^-$ current).

We calculated unitary aspartate transport rates from combined whole-cell current and fluorescence amplitudes of an earlier publication: in this dataset, whole-cell current amplitudes increase linearly with cellular expression levels quantified as whole-cell fluorescence, and we compared fluorescence-normalized current values from this publication in combination with its calculated glutamate transport of $561 \pm 123$ s$^{-1}$ [4]. Since aspartate currents exceed glutamate by a factor 2.3 at -160 mV, without the protons exchanged with glutamate at a stoichiometry of 1:1, we obtained a transport rate of approximately 2581 s$^{-1}$.

VGLUT1 structurally resembles transporters of the Major Facilitator Superfamily [30] and operates as a secondary active glutamate transporter. Therefore, we described glutamate and aspartate transport with an alternating access mechanism that shuttles between inward-facing states that permit the binding of amino acids from the cytoplasm and outward-facing states that release amino acids to the vesicular lumen (Fig 10).

The model is based on multiple assumptions. Since the Cl$^-$ channel is described using two protonation sites and shows a pH dependence remarkably similar to VGLUT1 transport of glutamate and aspartate, we assumed that the same number of sites are protonated when the empty transporter returns to the inward-facing conformation. Substrates then bind to the doubly protonated transporter and cross the membrane either in this protonation state or after the release of one proton. Substrate translocation in the doubly protonated state results in uncoupled uniport, and singly protonated transitions in H$^+$-substrate exchange. To permit H$^+$ transport, transporter protonation and deprotonation are permitted in inward- and outward-facing conformations. To ensure stoichiometrically coupled H$^+$-glutamate exchange, H$^+$ fluxes that are not coupled to glutamate transfer need to be prevented because they enable passive H$^+$ flux. Therefore, for transporters with no substrate bound, we permitted only doubly protonated translocation.

Optimized parameters for the transporter, separated by bound glutamate and aspartate substrates, is provided in S3 Table. Cl$^-$ is only present in the external solution, and we therefore exclusively accounted for Cl$^-$ binding in the outward-facing conformation. Since we studied glutamate and aspartate currents at negative voltages in the absence of

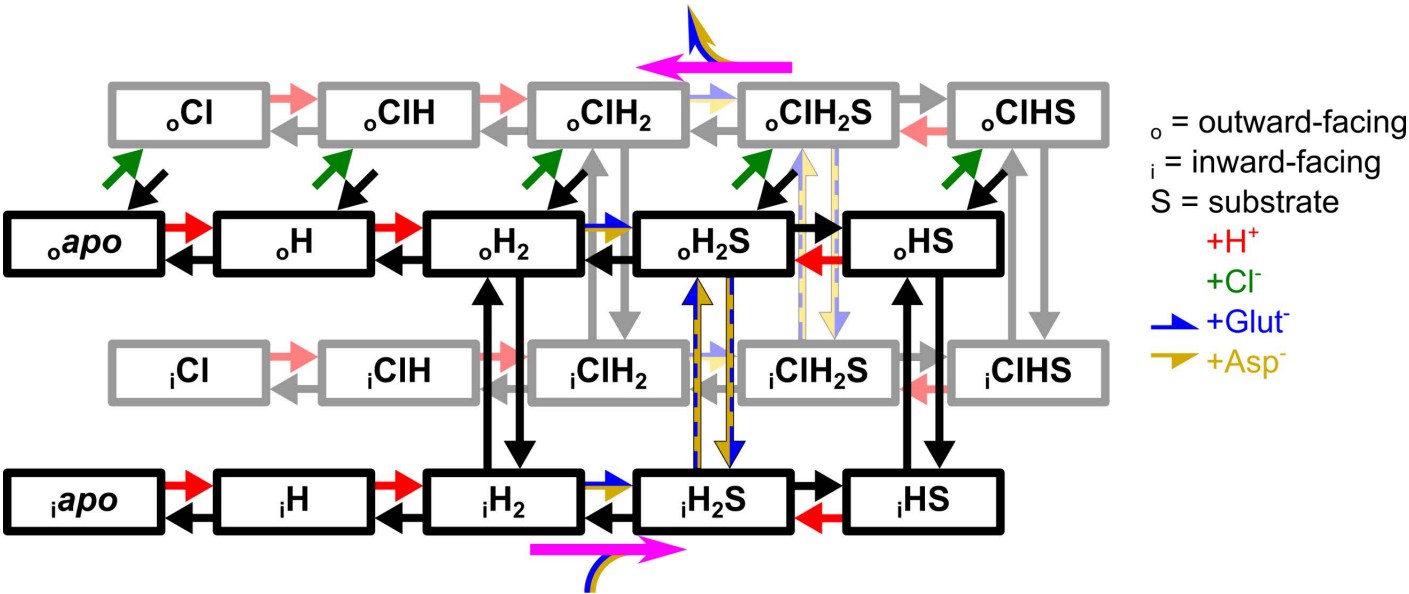

**Fig 10. Kinetic model of active glutamate and aspartate transport by VGLUT1.** The top states are outward-facing states at the bottom by vertical arrows. Green arrows depict Cl⁻ binding to the (outward-facing) black states in front, red arrows depict protonation, substrate binding is shown in horizontal blue-yellow arrows, and ligand unbinding in corresponding black arrows. Vertical black arrows with a blue-yellow fill show doubly protonated translocation with substrate, with limited rates when glutamate-bound. Pink arrows indicate the general direction of the Cl⁻-free (black states) and Cl⁻-bound (gray states) transport cycle.

internal Cl⁻, anion-conducting states were not added to the kinetic scheme. The scheme correctly describes the current kinetics and pH and [Cl⁻] dependence of both glutamate and aspartate currents (Fig 11), with both H⁺-glutamate and aspartate transport rates similar to experimental data [4].

**Cl⁻ stimulates cytoplasmic glutamate binding**

Fig 12 compares the kinetic parameters for coupled glutamate transport in the Cl⁻-free and Cl⁻-bound states; its voltage dependence is given by the corresponding $z$ and $d$ parameters in S9 Fig. VGLUT1 binds substrates from the internal solution after double protonation and translocation to the inward-facing state. At -160 mV, external Cl⁻ enhances the rates of glutamate binding to the inward-facing state from $6.9 \times 10^4$ ($3.5 \times 10^4$–$9.1 \times 10^4$, median and parameter amplitude range) $M^{-1} s^{-1}$ to $6.0 \times 10^6$ ($2.8 \times 10^6$–$2.4 \times 10^7$) $M^{-1} s^{-1}$ (Fig 12B). Changing the glutamate binding rate while modifying unbinding to maintain the same $K_D$ results in sharply defined and distinct RSS values in both conformations, which shows that Cl⁻ increases individual association rates and not just the $K_D$ (S10 Fig). This change in association rates suggests that the pocket needs to be primed upon Cl⁻ binding to permit binding of glutamate. Cl⁻ does not affect predicted $pK_a$ values for protonation after glutamate binding, they are close to pH 1 without Cl⁻ at 1.7 (0.6–2.1) or with Cl⁻ at 0.3 (-0.6–1.3). These low $pK_a$ values ensure deprotonation of the inward-facing glutamate-bound transporter at neutral cytoplasmic pH (Fig 12C).

Outward translocation rates of transporters with glutamate bound as substrate (ᵢHS) did not differ with and without Cl⁻ (Fig 12D), and neither did the $pK_a$ of the subsequent protonation (Fig 12E). Glutamate release to the lumen was impaired by Cl⁻, from $3.1 \times 10^{10}$ ($1.5 \times 10^{10}$–$8.1 \times 10^{10}$) $s^{-1}$ to $7.4 \times 10^8$ ($4.4 \times 10^8$–$3.1 \times 10^9$) $s^{-1}$ (Fig 12F). With a small increase in the final substrate-free (ₒH₂) inward translocation step from 3152 (2608–6165) to 9293 (7602–9966) $s^{-1}$ (Fig 12G), Cl⁻ provides a net positive effect on glutamate transport. We set an upper limit of $1 s^{-1}$ to the translocation rate constant of the doubly protonated substrate-bound transporter (ₒH₂S) to the inward-facing state only with glutamate as the substrate (Fig 12H).

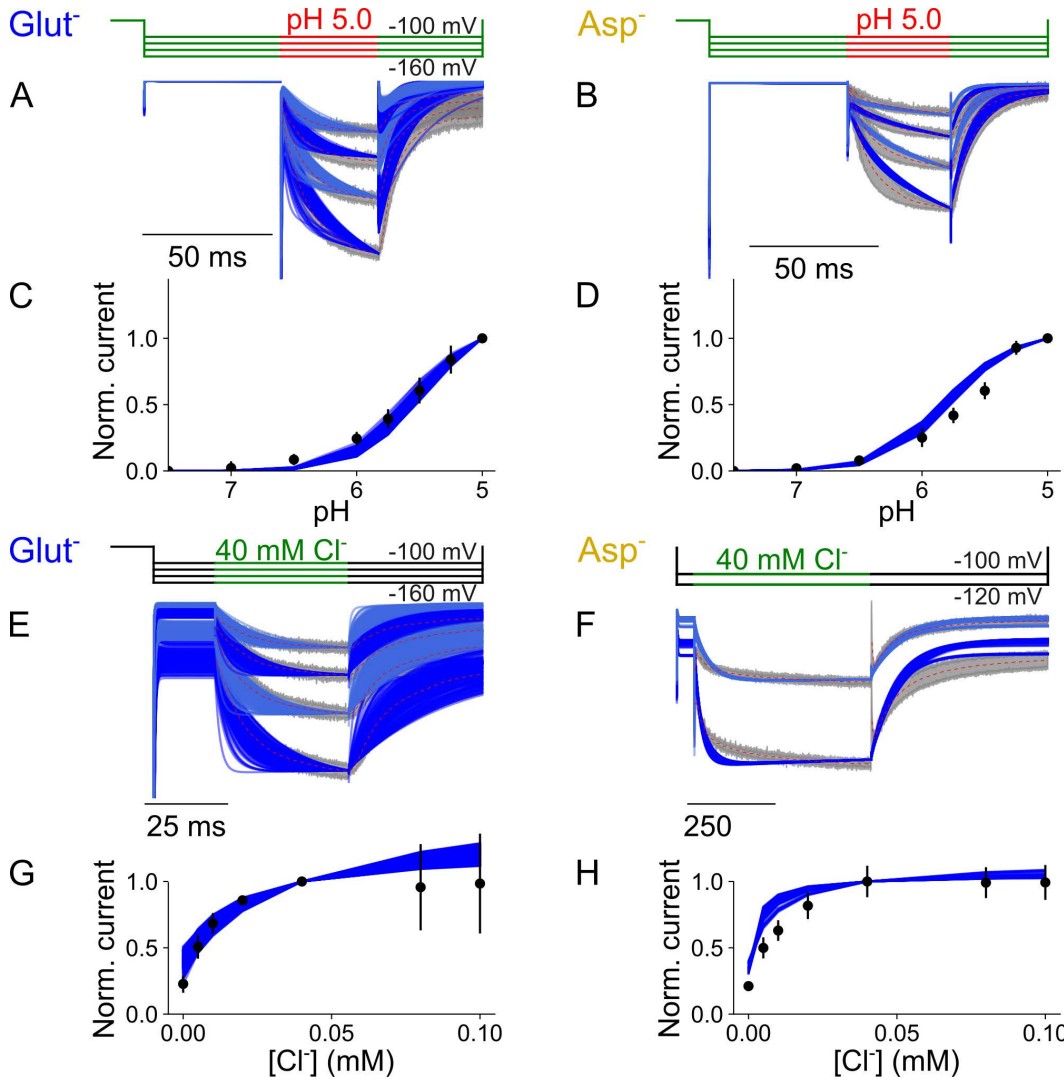

**Fig 11. VGLUT1$_{PM}$ glutamate and aspartate currents are well described by an alternating access kinetic scheme. (A, B)** Experimental currents and predictions by the kinetic scheme for pH steps from 7.4 to 5.0 at $[Cl^-]_o = 140\,mM$ with glutamate (A) or aspartate **(B)**. **(C, D)** Plots showing late currents (means ± 95% confidence interval) against pH for glutamate (C) or aspartate **(D)**. **(E, F)** Experimental currents and predictions by the kinetic scheme for changes in $[Cl^-]_o$ from 0 to 40 mM at pH 5.0 for glutamate (E) or at pH 5.5 for aspartate **(F)**. **(G, H)** Plots showing late currents (means ± 95% confidence interval) against pH at $[Cl^-]_o = 40\,mM$ for glutamate (G) or aspartate **(H)**. All panels show experimental data as the 95% confidence interval of patch clamp data (gray and black) and the fitting results of 250 simulations from parameter sets generated through exploratory mutations (blue) under the same conditions.

Glutamate transport is pronouncedly voltage dependent. The comparison of kinetic parameters at 0 mV and -160 mV in the presence of Cl⁻ demonstrate that the main effect of voltage is stimulation of the protonation of the substrate bound transporter (Fig 12E).

## Allosteric regulation of transporter protonation contributes to VGLUT1 substrate selectivity

Fig 13 compares the p$K_a$ values and transition rates for glutamate and aspartate transport in the Cl⁻-bound cycle, its voltage dependence is given by the corresponding $z$ and $d$ parameters in S11 Fig. Glutamate and aspartate bind at

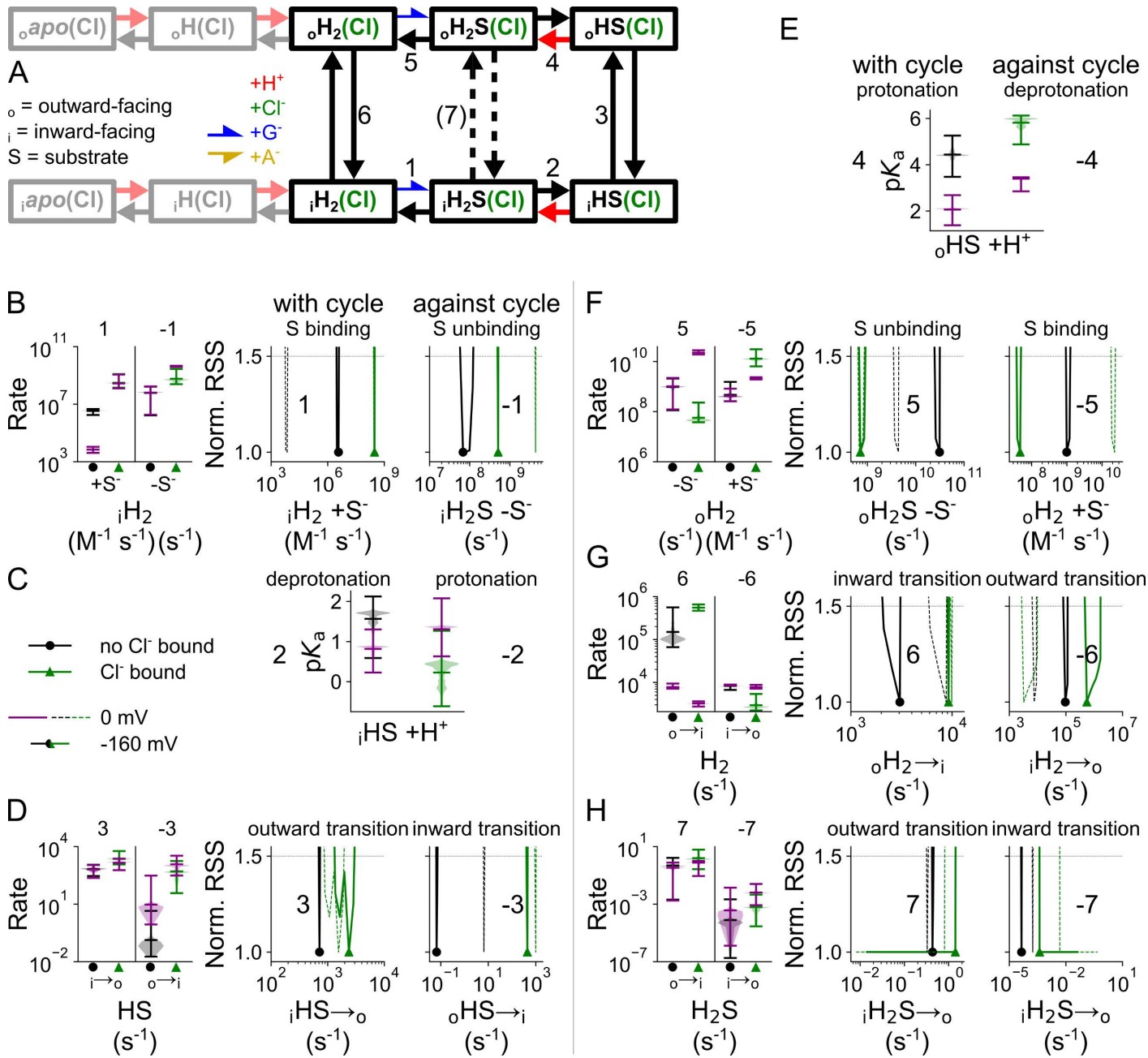

**Fig 12. Cl⁻ binding changes VGLUT1 glutamate transport parameters. (A)** Kinetic scheme describing glutamate transport by VGLUT1 with or without bound Cl⁻. **(B–H)** Cl⁻-induced changes in glutamate binding/unbinding in the inward-facing conformation **(B)**, in the p$K_a$ of the second protonation site in the inward-facing conformation **(C)**, in the singly protonated translocation rate **(D)**, in p$K_a$ for the second protonation site in the outward-facing conformation **(E)**, in glutamate binding/unbinding in the outward-facing conformation **(F)**, and in the doubly protonated translocation rate without substrate **(G)** or doubly protonated translocation rate with substrate **(H)**. Rates along the transport cycle are positively numbered and on the left, next to rates going against the transport cycle with negative numbers. Protonation is represented by p$K_a$ violin plots; other simulated rates are given as normalized RSS representing goodness of fit for a range of amplitudes in addition to violin plots depicting the amplitude range generated by exploratory mutation. All rates are shown at -160 mV or at 0 V in purple or dashed lines.

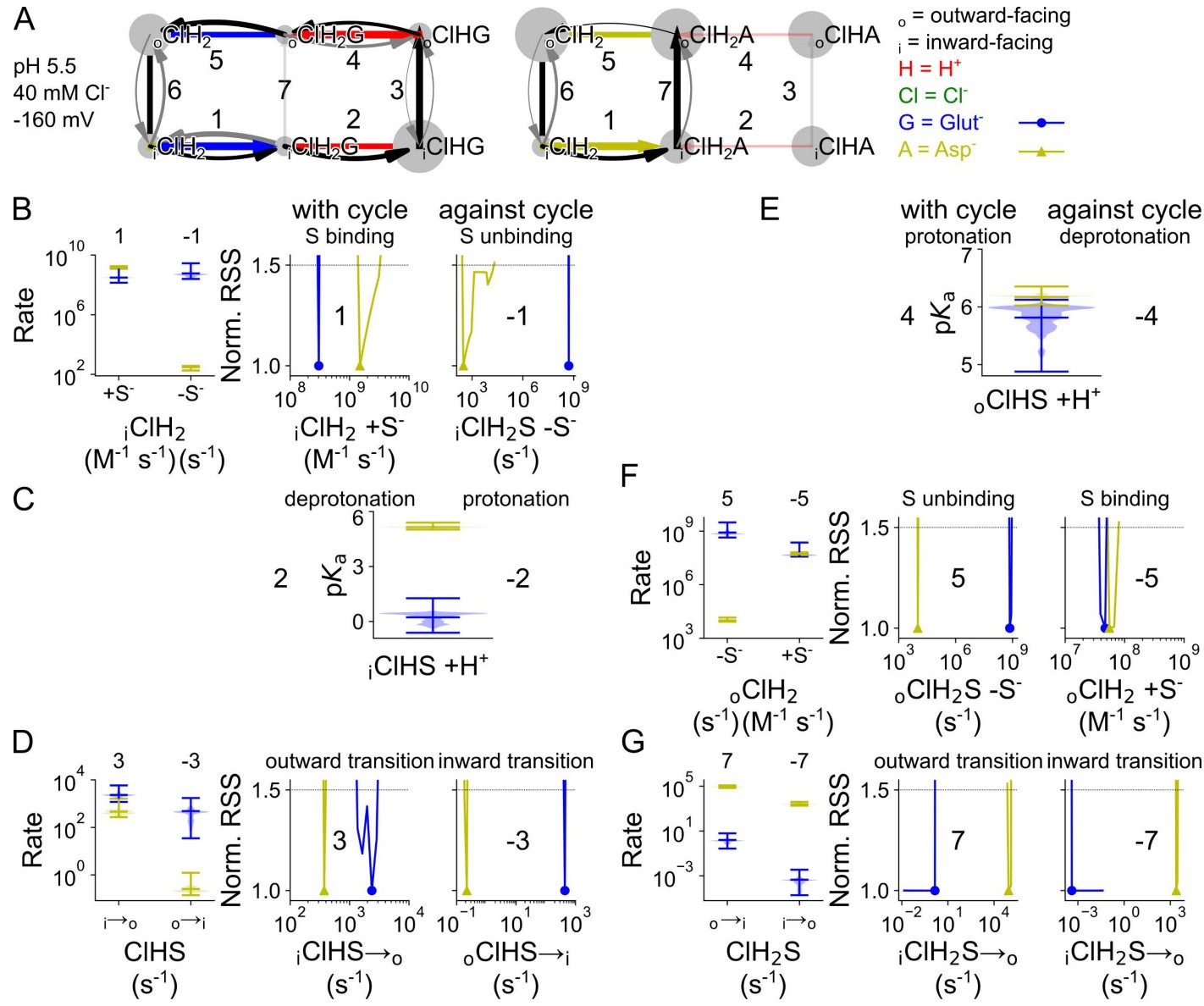

**Fig 13. Differences in the VGLUT1 glutamate and aspartate transport cycle. (A)** Kinetic scheme describing Cl⁻-bound and protonated VGLUT1 glutamate (left) or aspartate (right) transport function. (B) rates of substrate binding/unbinding in the inward-facing conformation. (C) p$K_a$ values for the second protonation site in the inward-facing conformation. **(D)** Translocation rates for the singly protonated, substrate-bound conformation. (E) p$K_a$ values for the second protonation site in the outward-facing conformation. **(F)** Rates of substrate binding/unbinding in the outward-facing conformation. **(G)** Translocation rates for the doubly protonated, substrate-bound conformation. Protonation is described by p$K_a$ violin plots; other simulated rates are given as normalized RSS representing goodness of fit for a range of amplitudes in addition to violin plots depicting the amplitude range generated by exploratory mutation. Data for glutamate transport are shown in blue and for aspartate transport in yellow. All rates are shown at -160 mV.

comparable rates to the doubly protonated inward-facing transporter, but with $5.1 \times 10^8$ ($2.5 \times 10^8$–$2.8 \times 10^9$, median and parameter amplitude range at -160 mV) s⁻¹ glutamate unbinding is significantly faster than aspartate unbinding with 309 (181–366) s⁻¹ (Fig 13A and 13B). The p$K_a$ of the second protonation site is 0.3 (-0.6–1.3) for glutamate-bound transporters and 5.1 (5.0–5.4) for aspartate-bound transporters (Fig 13C).

This difference results in preferred deprotonation of the glutamate-bound than of the aspartate-bound transporter; it supports glutamate translocation in the singly protonated state and $H^+$-glutamate exchange. We tested the importance of the different substrate-bound $pK_a$ values by modifying unbinding rates in both inward- and outward-facing conformation. Increasing the low glutamate value and decreasing the high aspartate value reduces transport rates, showing how important the optimized rates are for VGLUT1 transport rates (S12 Fig).

Substrate-bound outward translocation in the singly protonated state is the same for glutamate and aspartate (Fig 13D). Outward-facing $pK_a$ values with aspartate were the same as the glutamate-bound transporter (Fig 13E), at a value comparable to the inward-facing aspartate-bound $pK_a$. With $7.4 \times 10^8$ ($4.4 \times 10^8$–$3.1 \times 10^9$) $s^{-1}$, outward-facing VGLUT1 releases glutamate faster than aspartate with $1.0 \times 10^4$ ($8.4 \times 10^3$–$1.4 \times 10^4$) $s^{-1}$ with similar binding rates (Figs 13F and S12 Fig).

To ensure coupled $H^+$-glutamate transport, the glutamate translocation rate constant was limited to $1\,s^{-1}$ for doubly protonated VGLUT1. Under these restrictions, translocation is much faster for aspartate with $8.5 \times 10^4$ ($6.9 \times 10^4$–$1.2 \times 10^5$) $s^{-1}$ than for glutamate with $1.4$ ($0.3$–$6.2$) $s^{-1}$ while singly protonated outward translocation rates are equivalent (Fig 13D). This is consistent with the larger aspartate currents at very negative voltages in whole-cell recordings [4].

### Aspartate binding induces an occluded state that impairs Cl⁻ binding/unbinding

$VGLUT1_{PM}$ binds Cl⁻ in different protonation and substrate-bound states (Fig 14, the corresponding $z$ and $d$ parameters are given in S14 Fig). In transporters with no substrate bound, Cl⁻ binding has more pronounced effects on protonation of inward-facing states than of outward-facing states (Fig 14A). For inward-facing conformations, Cl⁻ decreases the $pK_a$ of the first protonation from $6.3$ ($6.0$–$6.6$) to $0.5$ ($0.4$–$0.7$), but increases the $pK_a$ of the second protonation from $4.0$ ($3.8$–$4.0$) to $7.3$ ($7.2$–$7.5$). For outward-facing conformations, Cl⁻ slightly increases $pK_a$ values for the first and second protonation states, from $7.8$ ($7.7$–$7.9$) to $8.2$ ($8.1$–$8.4$) and from $7.7$ ($7.5$–$7.9$) to $8.2$ ($8.1$–$8.4$), respectively. Both the $pK_a$ values and their Cl⁻ dependence are different from those of the protonation sites involved in VGLUT1 anion channel opening (Fig 4), consistent with the notion that different protonation sites are used for channel activation and transporter translocation.

Cl⁻ unbinding from transporters without substrate is promoted by deprotonation, despite a simultaneous but smaller increase in binding. With $0.48$ ($0.35$–$0.67$) mM Cl⁻ the non-protonated state has the highest $K_D$, with a lower values of $0.18$ ($0.12$–$0.28$) mM in the singly protonated state and $0.051$ ($0.038$–$0.076$) mM in the doubly protonated state (Fig 14C). However, statistical testing only identified significant differences in individual Cl⁻ association rates for substrate-bound transporters in the singly protonated state ($_oHS$; S15 Fig). For substrate-bound VGLUT1 in the single protonation state, Cl⁻ binding and unbinding is much slower with bound aspartate than with bound glutamate: Cl⁻ binding with $1.1 \times 10^4$ ($7.7 \times 10^2$–$2.2 \times 10^4$) $M^{-1}\,s^{-1}$ versus $5.6 \times 10^5$ ($6.8 \times 10^4$–$4.6 \times 10^6$) $M^{-1}\,s^{-1}$ and Cl⁻ unbinding at $0.529$ ($0.002$–$1.433$) $s^{-1}$ versus $1.9 \times 10^5$ ($1.7 \times 10^4$–$3.1 \times 10^6$) $s^{-1}$ (Fig 14D). For double protonation, the Cl⁻ binding/unbinding rates for the glutamate- and aspartate-bound states are much closer; however, aspartate binding results in tighter Cl⁻ binding with a $K_D$ of $9.5 \times 10^{-8}$ ($7.2 \times 10^{-8}$–$1.6 \times 10^{-7}$) mM compared to glutamate with a $K_D$ of $0.024$ ($0.003$–$0.078$) mM (Fig 14D). These changes are directly observable in fast Cl⁻ application experiments (Fig 9F), and indicate that binding of aspartate, but not of glutamate, modifies the Cl⁻ binding site towards higher affinity and lower association rates.

### Discussion

Sustained synaptic activity requires the rapid recycling and refilling of synaptic vesicles. In glutamatergic synapses, VGLUTs fulfill two major refilling tasks: they accumulate glutamate and—to ensure that vesicular filling is electrically and osmotically neutral—mediate the efflux of Cl⁻, which is present at a high concentration in vesicles after endocytosis [1,4,5,7]. Both transport functions are regulated by luminal [Cl⁻] [8–11,13], and we studied the kinetic basis for this. We found that Cl⁻ accelerates $H^+$-glutamate exchange, mainly by making the glutamate-binding site accessible to the cytoplasm, and by stimulating the inward translocation after substrate release to the vesicular lumen (Fig 12). Moreover,

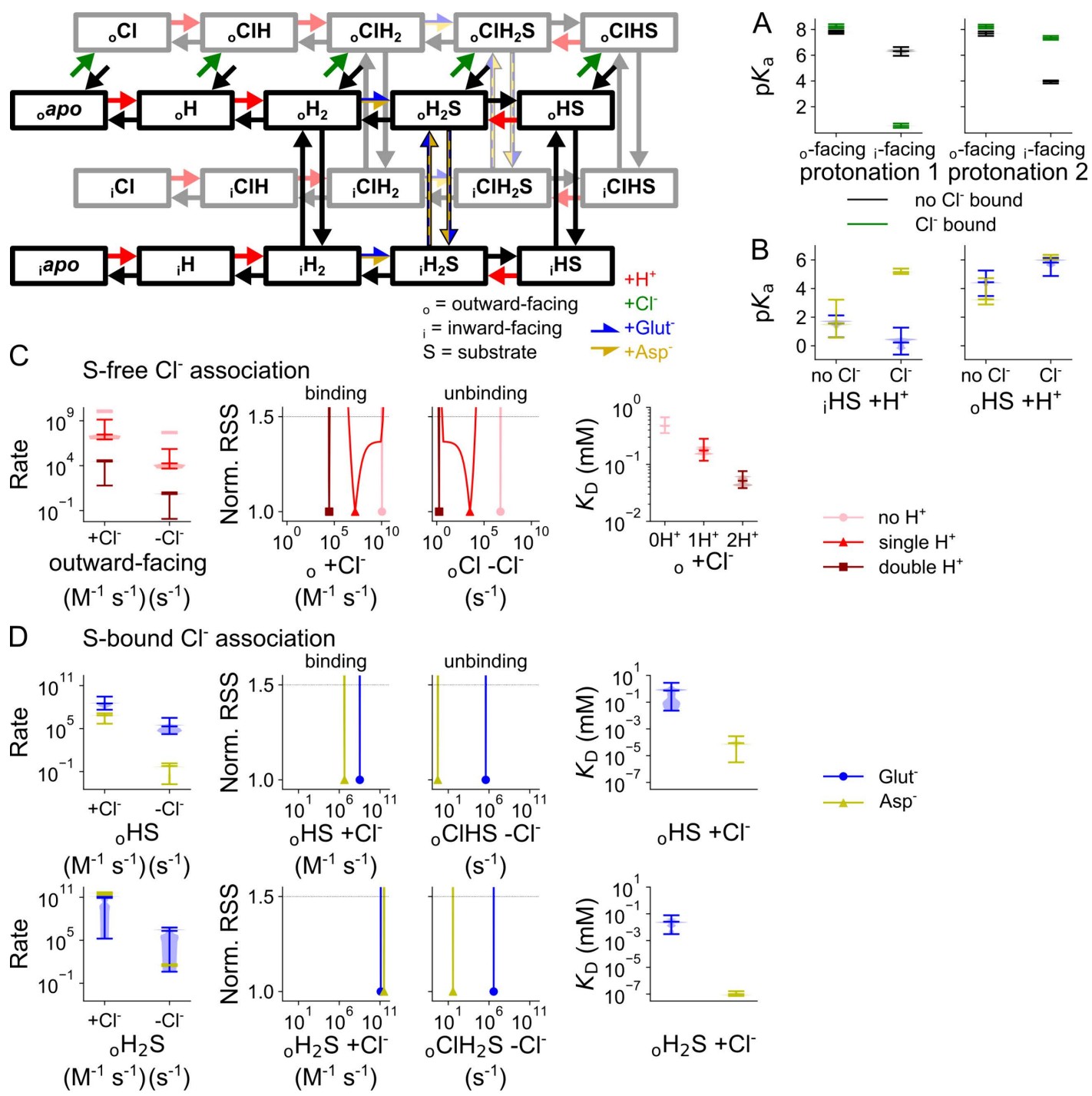

**Fig 14. Binding and unbinding of protons and Cl⁻ to VGLUT1 in its neurotransmitter transport mode.** (A) p$K_a$ for the first and second protonation, with and without Cl⁻, for inward- and outward-facing transporters without substrate. (B) p$K_a$ of the second protonation with bound substrate, with or without bound Cl⁻. **(C)** Cl⁻binding/unbinding rates and corresponding $K_D$ values for outward-facing transporters without substrate (light→dark red indicates increasing protonation). **(D)** Cl⁻binding/unbinding rates and corresponding $K_D$ values for the singly or doubly protonated, substrate-bound, outward-facing state). Protonation is represented by p$K_a$ violin plots; other simulated rates are given as normalized RSS representing goodness of fit for a range of amplitudes in addition to violin plots depicting the amplitude range generated by exploratory mutation. All rates are shown at -160 mV.

Cl⁻ increases the p$K_a$ of the second protonation site of the aspartate-bound inward-facing transporter, thereby promoting double protonation and preventing H⁺-coupling of aspartate transport (Fig 14B). During the activation of VGLUT1 anion channels, Cl⁻ stimulates the first protonation of the closed channel, accelerates opening of the singly protonated closed channel, and then stabilizes the open state by virtually abolishing channel closure regardless of protonation (Fig 4). The H120A point mutation modifies VGLUT1 Cl⁻ channel gating and anion conduction [4]. We found this mutation to increase the H⁺-binding affinity of the *apo* state well above to the levels observed in the WT, especially in the absence of Cl⁻, and to impair Cl⁻ binding by reducing its binding to the closed channel and increasing unbinding from the open channel for all protonation states (Fig 8). Therefore, our results identified the mechanistic basis of the allosteric control of VGLUT1 function by Cl⁻ and confirmed the functional importance of this regulatory mode.

We studied VGLUT1 anion channel activity in the absence of glutamate [4,12] and neurotransmitter transport after the complete substitution of intracellular anions with glutamate/aspartate at negative voltages with negligible Cl⁻ influx. These conditions enabled us to separately analyze VGLUT1 anion channel function and glutamate/aspartate transport. We developed distinct kinetic schemes to describe the two sets of experiments: anion channel function was described with a kinetic scheme that includes three VGLUT1 protonation states, each with a closed and open anion channel conformation and all of which can bind Cl⁻ (Fig 2); glutamate and aspartate transport was described with an alternating access transport scheme (Fig 10). It is likely that VGLUT1 anion channels can also open within the glutamate transport cycle under physiological conditions. However, generating a kinetic scheme for the combined transport/channel cycle will require better structural insight into the formation of anion channels than currently available.

Both VGLUT1 transport functions depend on H⁺ and Cl⁻ association/dissociation rates that are beyond the experimental time resolution, and there are protonation steps with p$K_a$ values outside pH values tolerated in our experiments. Kinetic modeling permits extending these parameter ranges by fitting time courses and concentration dependences under various experimental conditions. For glutamate/aspartate transport as well as for anion channel function, we used the minimal models sufficient to describe the existing experimental data. Our kinetic models do not separately describe each proton binding/unbinding process or conformation transition but, instead, combine distinct processes for which experimental data do not permit separation. Since the gating of VGLUT1 anion channels needs to be described with at least two protonation states, we also assumed two H⁺-binding sites in the transporter mode. VGLUTs contain many titratable residues [31], and more than two of these might be protonated during VGLUT transport/channel functions. Protonation sites in the channel and transporter modes indeed differed in p$K_a$ values as well as in Cl⁻ modulation (Figs 6 and 14), indicating that different sites determine the pH dependence of these two functions in VGLUTs. Many titratable VGLUT residues project into water-filled vestibules [31], and electrostatic interactions with Cl⁻ or with negatively charged amino acids might modify their p$K_a$ values [32–34]. During the activation of VGLUT1 anion channels, Cl⁻ increases the p$K_a$ values of the first protonation site in the closed, but not in the open, conformation and increases the p$K_a$ values of the second protonated open, but not the corresponding closed conformation (Fig 4). These differences suggest that Cl⁻ binds near the first protonation site, when the VGLUT1 anion channel is in the closed conformation. For the second protonation step, Cl⁻ appears to bind at a nearby site in the open, but not the closed, state. Resolving the open channel conformation at the molecular level may help to identify the VGLUT protonation sites involved in anion channel activation. Cl⁻ has additional effects on the opening and closing rates of VGLUT1$_{PM}$ anion channels, and similar effects of permeating ions have already been described for many ion channels. Cl⁻ binding may affect hydrophobic gating or prevent channel closing by a steric interaction via a foot-in-the-door mechanism [35,36].

Kinetic modeling provided new mechanistic insight into VGLUT anion channel activation: at neutral luminal pH, none of the responsible protonation sites are protonated and opening rates are negligible. Furthermore, Cl⁻ association rates are close to the diffusion limit, consistent with the VGLUT2 outward-facing conformation in recent cryogenic electron microscopy (cryo-EM) findings [31]. Luminal Cl⁻ supports protonation of the first site in the closed conformation; however, this induces a conformational state with impaired Cl⁻ accessibility from the external medium (Fig 5). Thus, the predominant

anion channel activation pathway includes Cl⁻ binding to non-protonated VGLUT1$_{PM}$, followed by protonation, transition into an occluded state and, finally, channel opening. VGLUT1$_{PM}$ anion channels are voltage-dependent, and channel opening upon membrane hyperpolarization and channel closure upon depolarization form the basis for the strong rectification of VGLUT1$_{PM}$ Cl⁻ currents [4]. Membrane hyperpolarization decreases the p$K_a$ values for closed-state non-protonated channel by one or two units, but increases them for the second protonation (Fig 4A). Negative voltages promotes opening rates of singly protonated channels by 2–3 orders of magnitude, but do not increase the opening rates of the doubly protonated channel. Taken together, the results show that membrane hyperpolarization increases the open probability of the VGLUT1$_{PM}$ anion channel by promoting the opening rates of the singly protonated state as the primary opening pathway, despite hampering the initial protonation.

To describe VGLUT1$_{PM}$ glutamate/aspartate transport we used an alternating access scheme (Fig 10). VGLUT1$_{PM}$ can transport many different anions, but glutamate transport is uniquely coupled to H⁺ exchange [4]. To account for this behavior, we assumed that glutamate would be transported across the membrane in a distinct transporter protonation state compared with other substrates (Fig 10). Glutamate and aspartate can modify VGLUT1 protonation in a similar allosteric fashion to Cl⁻ (Figs 13 and 14). The low p$K_a$ of inward-facing glutamate-bound transporters promotes H⁺ release from the second protonation site; inward-facing transporters with bound Cl⁻ and aspartate exhibit a p$K_a$ of 5.1 (5.0–5.4), thus making double protonation more likely (Fig 14B). Since retranslocation after substrate release occurs in a doubly protonated conformation, the preferential double protonation of inward-facing aspartate-bound states is sufficient to ensure aspartate uniport. S13 Fig illustrates the impact of this particular protonation on glutamate and aspartate transport rates. Increasing the p$K_a$ without further optimizing other rates results in steep reduction of the predicted H⁺-glutamate exchange rates, and decreasing the p$K_a$ towards glutamate-bound values similarly decreases aspartate transport.

Structural analysis of VGLUT2 suggests that glutamate is coordinated by the arginine residues R88 and R322 [31]. Aspartate is shorter and cannot bind both side chains simultaneously. These differences in the binding pocket might affect the protonation of nearby residues and may explain why glutamate promotes deprotonation of the substrate-bound inward-facing conformation. However, without limiting glutamate translocation in the doubly protonated state, our model fails to predict exclusive translocation for the singly protonated glutamate-bound conformation (Fig 13G). Therefore, we conclude that additional mechanisms ensure stoichiometrically coupled exchange, possibly steric effects that prevent glutamate-bound translocation in the doubly protonated conformation. The much slower activation time course of aspartate currents by Cl⁻ (Fig 9) indicates that—during the aspartate transport cycle—Cl⁻ binding either promotes a conformational change or requires a conformational change other than that induced by Cl⁻ binding to the glutamate transporter. A likely scenario is that intermediate occluded states are more stable when aspartate is bound rather than glutamate.

Together with inorganic phosphate transporters, the lysosomal H⁺/sialic acid cotransporter sialin, and the vesicular nucleotide transporter VNUT, VGLUTs form the SLC17 family. Except for sialin (SLC17A5), all SLC17 transporters are regulated by Cl⁻ [37]. Thus far, the physiological importance of this type of regulation remains insufficiently understood. Juge et al. (2010) demonstrated that ketone bodies block VGLUTs by competing with Cl⁻ at the allosteric regulation site and suggested that this regulatory mode will reduce glutamate release and inhibit network activity under conditions of insufficient food supply and increased ketone body production. Allosteric Cl⁻ regulation might also restrict glutamate accumulation in synaptic vesicles under physiological conditions. Neurons exhibit intracellular [glutamate] of 5–10 mM [38], and VGLUT1-mediated electrogenic H⁺-glutamate exchange may generate equilibrium glutamate concentrations of above 1 M in synaptic vesicles, causing water influx and vesicle swelling. The dual function as anion channel and transporter, together with allosteric Cl⁻ regulation, may prevent such glutamate overloading: VGLUT chloride channels deplete Cl⁻ from synaptic vesicles during glutamate filling, and the resulting low luminal [Cl⁻] will block glutamate uptake at a certain filling level.

H120A VGLUT1 profoundly alters H⁺-glutamate coupling and anion channel gating [4], but kinetic modeling demonstrated only minor alterations in the kinetic scheme developed for WT VGLUT1 anion channels. The H120A mutation

promotes closed channel protonation, especially without Cl⁻, but reduces protonation under most other conditions. The mutation reduces the opening rates of singly protonated channels, while increasing doubly protonated opening when Cl⁻ is bound. Moreover it increases closing rates, and these changes together reduce the stability of the singly protonated channel. This may explain why it takes longer to reach steady state current. In the open channel with no Cl⁻ bound, the H120A mutation causes no meaningful changes. These results demonstrate that H120 is not one of the protonation sites necessary for anion channel opening. They rather indicate that Cl⁻ binding is impaired in virtually all protonation states of open and closed H120A VGLUT1$_{PM}$ channels (Fig 8C). Experimental [13] and computational [39] work identified R176/R184 (VGLUT1/VGLUT2) as the luminal Cl⁻-binding site residue. In atomistic simulations, protonated H128 (in VGLUT2, corresponding to H120 in VGLUT1) was shown to support Cl⁻ binding to R184 [39], in full support of our kinetic modeling results. VGLUTs differ from related transporters in a larger number of potential H⁺ acceptors [31,40,41] and residues that need to be protonated for anion channel opening, allosteric transporter activation or H⁺-glutamate exchange have not been unambiguously identified.

In summary, we have identified modification of H⁺ binding as a key process in VGLUT glutamate transport and anion channel activation. Transmembrane voltage promotes anion channel opening with single protonation, and luminal Cl⁻ increases the p$K_a$ values to reach this protonation state as well as stabilizing open channels in general (Fig 4). Differences in transporter protonation contribute to the distinct transport stoichiometries of glutamate or aspartate: glutamate binding promotes H⁺ release to the cytoplasm and permits preferential glutamate translocation in the singly protonated state as basis of stoichiometrically coupled H⁺-glutamate exchange, whereas aspartate binding does not (Fig 13C). We found that Cl⁻ promotes conformational changes, most importantly by facilitating substrate binding from the cytoplasm (S12 Fig) or by increasing anion channel opening and substantially decreasing channel closing rates (Fig 8B). Analyzing the rates of Cl⁻ binding/unbinding demonstrated that Cl⁻ association is impaired in certain protonation states during channel activation (Fig 5A), likely via the formation of occluded states or via translocation into inward-facing conformations. Taken together, our results illustrate how vesicular glutamate transporters use allosteric interactions between the anionic substrate glutamate, the main physiological anion Cl⁻, and the driving co-substrate H⁺ to orchestrate multiple functions of these key players in synaptic transmission.

## Materials and methods

### Expression plasmids, mutagenesis, and heterologous expression

HEK293T cells (Sigma-Aldrich) were grown in humidified incubators at 37°C and 95% air/5% $CO_2$ in Dulbecco's modified Eagle Medium supplemented with 10% FBS, and 5 mL penicillin-streptavidin at 5,000 U/ml. To express VGLUT1 as an eGFP fusion protein, we inserted the coding region of eGFP into pcDNA3.1-rVGLUT1 (kindly provided by Dr. Shigeo Takamori at Doshisha University in Kyoto, Japan) at the 3′ end of the transporter coding region and mutated various dileucine-like endocytosis motifs to alanine to promote plasma membrane insertion using PCR-based techniques [4]. The H120A mutation was introduced into VGLUT1$_{PM}$ using PCR-based techniques. Cells were transiently transfected using a calcium phosphate or lipofectamine precipitation method and examined 24 h later.

### Whole-cell patch clamp and fluorescence measurements

Standard whole-cell patch clamp recordings were performed using EPC10 amplifiers, controlled by HEKA PatchMaster (Multi Channel Systems MCS GmbH, Reutlingen, Germany; [4]. We used borosilicate pipettes (Harvard Apparatus, Holliston, Massachusetts, USA) with a resistance of 0.9–3 MΩ and applied series resistance compensation and capacitance cancelation, resulting in a voltage error of < 5 mV. The standard bath solution contained (in mM) 136 choline chloride, 2 $MgCl_2$, and 30 HEPES or 50 MES; we adjusted the osmolarity of the bath solution with glucose to values at least 5 mOsm/L higher than the internal solution. HEPES was used as the buffer for external and internal pH values between 6.5 and 7.4, and was replaced with MES for more acidic perfusion solutions; in all cases, pH was adjusted with choline

hydroxide. In experiments that used glutamate or aspartate in the pipette solution, 100 mM choline chloride was replaced by choline gluconate in the bath solution; cells were held at -50 mV before all measures other than the Cl⁻ current. The standard pipette solution contained (in mM) 140 choline anion$_X$, 5 EGTA, 5 Mg(OH)$_2$, and 30 HEPES, adjusted to pH 7.4 with TMA-OH (anion$_X$ = Cl⁻, glutamate⁻ or aspartate⁻). For glutamate- or aspartate-based pipette solutions, we used internal agar salt bridges (made from plastic tubing filled with 0.5 M KCl in 2% agar) to connect the Ag/AgCl electrode. Where necessary, junction potentials were calculated and corrected. To support complete intracellular dialysis, we waited at least 2 min after establishing the whole-cell mode before starting electrical recordings. Except for Cl⁻ dependence of the anion channel function, current recordings were corrected for leakage currents by subtracting current amplitudes obtained in the same cell with external pH 7.4 under identical conditions. In an earlier publication [4], we carefully tested whether glutamate and aspartate currents measured in this way are mediated by VGLUT1$_{PM}$: both are blocked by Rose Bengal (a high-affinity, non-competitive VGLUT blocker), only negligible currents can be recorded under these conditions in untransfected HEK293T cells, and there is a linear relationship between expression levels and transport currents.

Fast solution exchanges were achieved by moving the solution interface of continuous perfusion flow around a cell using a PiezoMove P-601.30L or P-840.40 piezoelectric actuator (Physik Instrumente, Karlsruhe, Germany) controlled via a MXPZT controller (Siskiyou, Grants Pass, Oregon, USA) or PI E-836 Piezo Driver combined with a USBPGF-S1 Instrumentation Amplifier Low Pass Filter (Alligator Technologies, Charlottesville, Virginia, USA) to smoothen the signal and reduce vibration. Actuators were connected to the EPC10 amplifier and controlled using HEKA PatchMaster software. The mean time course of solution exchange was determined to be 0.87 ± 0.2 ms (n = 19, mean and 95% confidence interval) by recording the current responses of an open pipette after changing from a high to low [Cl⁻] solution and measuring the time passing between 10% and 90% of the maximum current (S1 Fig). Two back-and-forth exchanges were performed, and the on-off pair with the best resolution was chosen and averaged. Even the slowest solution exchange is less than half of the fastest solution exchange time constant of our experimental data (S1C Fig).

### Kinetic modeling

VGLUT chloride or glutamate/aspartate currents were simulated using kinetic models based on subsequent conformational changes and ligand binding/reactions. Changes in state occupation were computed using a set of differential rate equations

$$\frac{dy(t)}{dt} = A \cdot y(t)$$

Eq 1

in which the state of the system is represented by the vector y(t), and transitions between states are encoded in a transition matrix, which contains the rate constants that determine the dynamics of the system. A simplified example of such states and matrix are shown in Fig 15. The evolution of the system is described by numerically integrating the differential

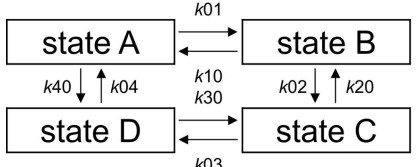

| | state A | state B | state C | state D |
|---|---|---|---|---|
| state A | -$k01$ -$k40$ | +$k10$ | | +$k04$ |
| state B | +$k01$ | -$k02$ -$k10$ | +$k20$ | |
| state C | | +$k02$ | -$k03$ -$k20$ | +$k30$ |
| state D | +$k40$ | | +$k03$ | -$k04$ -$k30$ |

**Fig 15. Hypothetical kinetic model and corresponding transition matrix.** Directional rate numbers are chosen arbitrarily and full matrix equations are omitted for clarity. Any ligand binding rate would additionally be multiplied by the concentration of k0X, according to the convention used, for both its appearances in the matrix.

equation, using the scipy.integrate.odeint function and providing the time evolution of the protein state distribution y(t). This method avoids conversions and lag time limitations by enabling time-continuous prediction of Cl⁻ ion fluxes and glutamate/aspartate transport. We only used time steps to compare the simulation output to experimental time course data in order to calculate RSS. These time steps were set to the recording frequency of the experiment, except the Cl⁻ jumps with aspartate current, which was filtered to mitigate its substantial length.

Solving this matrix as a series of linear algebra equations provided steady-state distributions of kinetic states. The dependence of net rates on voltage and temperature, as an adaptation of the Arrhenius equation, is calculated as

$$k01 = k1 \bullet e^{z \bullet d \bullet V \bullet \frac{F}{R \bullet T}}(\bullet [ligand])$$

Eq 2

where $k1$ is the rate constant (limited to $< 10^5$ s⁻¹ for conformation changes and to $5 \times 10^9$ for binding and unbinding of ligands), $z$ is charge movement (-1 to + 1), $d$ is a symmetry factor (0–1 [42], V is the voltage across the cell membrane, and the other elements are constants: $F$ is the Faraday constant, $R$ is the gas constant, and $T$ is the absolute temperature. The inverse reaction $k10$ was calculated as:

$$k10 = k2 \bullet e^{-z \bullet (1-d) \bullet V \bullet \frac{F}{R \bullet T}}$$

Eq 3

To ensure microscopic reversibility, one rate constant was calculated from the clockwise and counterclockwise products, and one $z$ variable was the sum of all other $z$ values in each cycle. To prevent transporter behavior that is incompatible with experimental data, the rate constants describing non-protonated channel opening and doubly protonated glutamate-bound transporter transition between inward- and outward-facing conformation, each with or without bound Cl⁻, were limited to 1 s⁻¹. The calculated net rates may exceed 1 s⁻¹ at -160 mV depending on their voltage dependence.

We used a population-based optimization procedure inspired by biological evolution, which was adapted from the Distributed Evolutionary Algorithms in Python (DEAP) software package [43]. To allow modification in either direction with minimal initial bias, we chose an initial parameter set with intermediate starting values of 1000 s⁻¹ for all rate constants, except the non-protonated channel opening and doubly protonated in- or outward transitions with glutamate, which were set to 1 s⁻¹ and proton-binding steps, which were set to $10^8$ s⁻¹. This initial parameter is copied into at least 50 parameter sets. With the exception of parameters calculated from others for detailed balance, all parameter sets were allowed to be mutated (p = 0.5) by a random amount. Additionally, randomly selected pairs of parameter sets may mate (p = 0.7); i.e., exchange values of specific parameters, with exchanged parameters being randomly chosen. Rate constants are limited to $10^5$ s⁻¹ for conformational changes and to $5 \times 10^9$ s⁻¹ for binding and unbinding of ligands. Charge movement can range from -1 to +1, and symmetry from 0 to 1. If mutations cause parameter values outside these bounds, individuals will be declared unfit and removed from the population. The fitness of each parameter set is evaluated as the cumulative difference between simulated and experimental data. Comparisons are made as residual sums of square for calculated values and experimental time courses. These residual sums of square are multiplied by a chosen weight factor – to adjust the category influence on fitness – and summed up. We usually increase weight multiplier for the worst fits, and scale down metrics that appear to dominate changes in fit quality. This repeated adjustment sustains a selective pressure to drive the optimization of simulation outputs where accuracy needs to be improved.

From a group of individuals, a tournament-style selection picks the parameter set with the best fitness: three parameter sets are randomly selected from the current generation, and the one with the best fitness is selected for each slot in the next generation (based on https://deap.readthedocs.io/en/devel/api/tools.html#deap.tools.selTournament).

We converted the kinetic transition rates into transition probabilities in order to allow for the calculation of preferred pathways of opening and closing of the anion channel (Figs 3L and 7L), using the reactive flux module of the Deeptime

Python library which uses transition path theory [44]. Activation pathways were described starting from the *apo* state to any conductive state, deactivation pathways from the two most common open states (oH or oClH$_2$) to either of the two unprotonated closed states (*apo* and cCl).

For the comparison of kinetic parameters, we used two tests, and changes were only considered significant after passing both tests. The first test is based on the collection of multiple parameter set variants generated via 'exploratory mutation' of a modified DEAP cycle. All parameter sets with RSS values that are less than 25% larger than the optimum values for all simulated metrics and time courses are shown in violin plots, together with the median and the confidence interval that provides the range of variation generated during exploratory mutation. Parameters are considered different when there is no overlap of the confidence interval. Whereas >10,000 sets with >3000 variants for each parameter were generated for the Cl$^-$ channel model, we limited the data size for glutamate/aspartate transport by removing individuals that did not vary parameters describing protonation and generated >980 variants from >20,000 individual sets from >50,000 generations. A second statistical test to calculate the change in the quality of fit is based on varying only the compared parameter [27]. In such a plot, a sharply defined and distinct parameter value is associated with instantly increasing RSS upon small changes in parameter values without overlap with parameter values obtained under other conditions. We assume two parameters to be indistinct when values overlap at RSS of below 150%.

For current predictions in the VGLUT1$_{PM}$ channel mode, the distribution of relative state occupations in the transition matrix was used to calculate the fraction of open states, i.e., the open probability, as the output value. Numerical weights were assigned to individual simulated metrics and time courses to control the driving force optimizing simulation accuracy through parameter mutation. A large weight ensures that the WT model remains close to the calculated open probability of 0.24 at 140 mM Cl$^-$, pH 5.5, and -160 mV [4]. Each simulation calculates the modification of steady-state currents by [Cl$^-$]$_o$, pH, or voltage relative to this open probability and the time-dependent changes of currents relative to the highest steady-state current in each experiment. Active transport was calculated as the sum of net transition fluxes between states, multiplied by the corresponding $z$ value. Here, a microscopic reversibility charge offset of -1 or 2 in each transport cycle accounts for current generation. Similarly to how Cl$^-$ channel models were trained to mimic experimentally determined open probability, the active transport model uses the calculated glutamate (561 s$^{-1}$) and aspartate (2581 s$^{-1}$) transport rates.

### Quantification and statistical analysis

Data are given either as the mean ± 95% confidence interval or as violin plots showing the distribution of data with the median and exploratory mutation value range. We calculated secondary binding rates for Cl$^-$, glutamate, and aspartate to normalize them, dividing the binding rate constant by the ligand concentration: in mM when calculating $K_D$ (expressed in mM) and in M for the rates themselves (expressed in M$^{-1}$ s$^{-1}$).

Measurements were taken from distinct samples. Mean and confidence intervals of time courses, dependence fits, and fit parameters were either from individual fits (chloride dependence of glutamate and aspartate) or determined by bootstrapping (other dependence fits and time constants). For patch clamp electrophysiology experiments, sample sizes correspond to the number of patched cells and are given in the Figure or legend. For statistical analysis of two groups, we used two-tailed bootstrap hypothesis testing via the bootstrap t-statistic of bootstrapped means, F-tests to determine the number of exponents in current activation and deactivation, and one-sample Wilcoxon signed rank tests when comparing fits of means to distributions of groups of individually fitted cells.

We fitted time courses of VGLUT1$_{PM}$ currents with mono- or biexponential functions and concentration dependence by global bootstrapping, with the Michaelis-Menten equation for Cl$^-$ and the Hill equation for pH to better capture saturation. To generate modest and comparable confidence intervals, we used sampling of 1000 for dependence data and time constants. To ensure reliable p-values, we used higher sampling of 10,000 for bootstrap hypothesis testing between different datasets.

# Supporting information

**S1 Fig. Extension of Fig 1. Validation of fast solution exchange experiments.** (A) Mean $Cl^-$ current responses to pH jumps for 5 untransfected HEK293T cells (yellow), in comparison to a representative HEK293T cell expressing VGLUT1$_{PM}$ under identical conditions (black). The pH jumps are from pH 7.4 to 5.5 and back, at -160–60 mV and with 140 mM external $Cl^-$. (B) Representative recording of the protocol used to test open pipette solution exchange time, typically between different [$Cl^-$]. (C) Calculated exchange times collected across several years compared to the fastest rate of all solution exchange experiments.
(TIF)

**S2 Fig. Extension of Fig 1. Time constants for VGLUT1$_{PM}$ $Cl^-$ current activation/deactivation in response to $H^+$ or $Cl^-$ concentration steps.** (A) activation/deactivation time constants upon pH jumps from 7.4 to 5.0 or 5.5 (left) or from 5.0 or 5.5 to 7.4 (right) at an external [$Cl^-$] of 0 or 140 mM. (B) time constants upon [$Cl^-$] steps from 0 to 40 mM or 0–140 mM at pH 5.5. Activation at high external $Cl^-$ or deactivation without $Cl^-$ were fitted with biexponential functions, providing two time constants and the relative amplitude (dashed lines) of the slower component. Data are shown as means obtained by bootstrapping with a global fit of experimental data with a sampling of 1000, with 95% of the sampling as error bars. Voltage differences are the result of *a posteriori* liquid junction potential correction.
(TIF)

**S3 Fig. Extension of Fig 4. Voltage dependence of parameters of the WT VGLUT1$_{PM}$ anion channel kinetic scheme.** (A) distribution of *z* and *d* parameters for protonation steps with and without $Cl^-$. (B) distribution of *z* and *d* parameters for channel opening. Protonation parameters are represented by violin plots, other simulation results are given as normalized RSS representing goodness of fit for a range of amplitudes in addition to violin plots depicting the amplitude range generated by exploratory mutation.
(TIF)

**S4 Fig. Extension of Fig 5. Voltage dependence of $Cl^-$ association parameters of the WT VGLUT1$_{PM}$ anion channel kinetic scheme.** Distribution of *z* and *d* parameters for the three protonation states (light→dark red indicates increasing protonation). Simulation results are given as violin plots depicting the amplitude range generated by exploratory mutation and normalized RSS representing goodness of fit for a range of amplitudes.
(TIF)

**S5 Fig. Extension of Fig 6. Comparison of WT and H120A VGLUT1 anion current kinetics.** (A) representative current responses to concentration jumps from pH 7.4 to 5.0 at [$Cl^-$] = 0 mM. (B) corresponding time constants for activation (left) and deactivation (right). (C) representative current responses to concentration jumps from pH 7.4 to 5.0 at [$Cl^-$] = 0 mM140 mM. (D) corresponding time constants for activation (left) and deactivation (right). (E) representative current responses to concentration jumps from [$Cl^-$] = 0–140 mM at pH 5.5. (F) corresponding time constants for activation (left) and deactivation (right) at given holding potentials. Activation at high external $Cl^-$ or deactivation without $Cl^-$ were fitted with biexponential functions, with two time constants and the relative amplitudes (dashed lines) of the slower component shown. Data are shown as means obtained by bootstrapping with a global fit of experimental data with a sampling of 1000, with 95% of the sampling as error bars. Voltage differences are the result of *a posteriori* liquid junction potential correction.
(TIF)

**S6 Fig. Extension of Fig 8. Voltage dependence of kinetic parameters of the H120A VGLUT1$_{PM}$ anion channel kinetic scheme.** (A) distribution of *z* and *d* parameters for protonation steps with and without $Cl^-$. (B) distribution for channel opening with and without $Cl^-$. (C) distribution for $Cl^-$ binding by protonation state. Protonation parameters are

represented by violin plots, other simulation results are given as normalized RSS representing goodness of fit for a range of amplitudes in addition to violin plots depicting the amplitude range generated by exploratory mutation.
(TIF)

**S7 Fig. Extension of Fig 8. Statistical analysis of Cl$^-$ binding by WT and H120A VGLUT1$_{PM}$ at a constant $K_D$.** Changes in the goodness of fit upon the modification of secondary Cl$^-$-binding rate constants at -160 mV, to closed (left) or open (right) WT or H120A VGLUT1$_{PM}$ anion channels in the unprotonated or singly or doubly protonated state. During modification, unbinding constants were simultaneously altered to keep the Cl$^-$-binding affinity unchanged.
(TIF)

**S8 Fig. Extension of Fig 9. Time constants for VGLUT1$_{PM}$ glutamate or aspartate current activation/deactivation by H$^+$ or Cl$^-$.** Activation/deactivation time constants upon pH jumps from 7.4 to 5.0 (left) or deactivation time constants upon pH jumps from 5.0 to 7.4 (right) at an external [Cl$^-$] of 40 mM (**A**) or upon [Cl$^-$] jumps from 0 to 40 mM (left) or deactivation time constants upon [Cl$^-$] jumps from 40 to 0 mM (right) at an external pH of 5.0 for glutamate or 5.5 for aspartate (**B**). Activation and deactivation of aspartate currents by Cl$^-$ were fitted with biexponential functions, providing two time constants and relative amplitudes (dashed lines) for the slower component. Data are shown as means obtained by bootstrapping with a global fit of experimental data with a sampling of 1000, with 95% of the sampling as error bars.
(TIF)

**S9 Fig. Extension of Fig 12. Voltage dependence of parameters of the glutamate transport cycles.** Distribution of $z$ and $d$ parameters for steps describing substrate binding ([1] and [5]), protonation ([2] and [4]), and transition between inward- and outward-facing conformations [3,6,7]. Simulation results are given as violin plots depicting the amplitude range generated by exploratory mutation and normalized RSS representing goodness of fit for a range of amplitudes.
(TIF)

**S10 Fig. Extension of Fig 12. Statistical analysis of glutamate binding rates, assuming a constant $K_D$. A,** Changes in the goodness of fit upon modification of the binding rate for glutamate with and without Cl$^-$, in the inward- and outward-facing conformation. During modification, the unbinding constants were simultaneously altered to keep the ratio (i.e., glutamate/aspartate-binding affinity) constant. Amplitudes are the optimized value plus 50 logarithmically distributed points between 1 and the ligand binding limit of $5 \times 10^9$; binding rates are at -160 mV and normalized to 140 mM glutamate or aspartate. The same RSS was derived from a wide range of rate values, indicating that the $K_D$ determines the RSS and that individual binding/unbinding rate constants do not play a major role.
(TIF)

**S11 Fig. Extension of Fig 13. Voltage dependence of protonation and substrate binding in the Cl$^-$-bound active transport cycles.** Distribution of $z$ and $d$ parameters for substrate binding (1 and 5), protonation steps (2 and 4), and transition between inward- and outward-facing conformations (3 and 7). Simulation results are given as violin plots depicting the amplitude range generated by exploratory mutation and normalized RSS representing goodness of fit for a range of amplitudes. Data for the outward transition of transport cycle step 6 (Fig 13) was omitted due to being substrate-independent.
(TIF)

**S12 Fig. Extension of Fig 13. Ligand-bound deprotonation rates are the major determinants of transport rates.** (A) predicted glutamate (blue) and aspartate (yellow) unitary currents (given as the number of charges per second in steady state × elementary charge) upon p$K_a$ modification. (B) relative number of H$^+$-substrate exchange transport cycles upon p$K_a$ modification. The optimized p$K_a$ value for the second protonation in the inward- or outward-facing substrate-bound

conformation is increased or decreased via the deprotonation rate constants by a factor of 3, 10, 33, or 100 while maintaining microscopic reversibility. Whereas the transport rates strongly depend on the $pK_a$ for both substrates, glutamate is transported in an exchange mode and aspartate in a uniport mode for all tested $pK_a$ values.
(TIF)

**S13 Fig. Extension of Fig 13. Statistical analysis for substrate binding rates, assuming constant $K_D$.** Statistical analysis for substrate binding rates, assuming constant $K_D$. Changes in the goodness of fit upon modification of the binding rate for glutamate and aspartate, with and without Cl⁻, in both conformations. During modification unbinding constants were simultaneously altered to keep the ratio; i.e., the glutamate/aspartate binding affinity, unaltered. Amplitudes are the optimized value plus 50 logarithmically distributed points between 1 and the ligand binding limit of $5 \times 10^9$, binding rates are at -160 mV and normalized to 140 mM glutamate or aspartate. The same RSS caused by a wide range of rate values indicates the $K_D$ determines the RSS, while individual binding/unbinding rate constants play no major role.
(TIF)

**S14 Fig. Extension of Fig 14. Voltage dependence of protonation and Cl⁻ binding in the VGLUT1$_{PM}$ active transport model.** (A) parameters for the first and second protonation without bound glutamate or aspartate for inward- and outward-facing conformations, as modulated by external Cl⁻. (B) parameters for the first and second protonation with substrate bound for inward- and outward-facing conformations. (C) parameters for Cl⁻ binding with substrate bound and with single or double protonation. (D) parameters for Cl⁻ binding with no substrate bound and with no, single, or double protonation. Protonation parameters are represented by violin plots, other simulation results are given as normalized RSS representing goodness of fit for a range of amplitudes in addition to violin plots depicting the amplitude range generated by exploratory mutation.
(TIF)

**S15 Fig. Extension of Fig 14. Statistical analysis for Cl⁻-binding rates, assuming a constant $K_D$.** Changes in the goodness of fit upon modification of the Cl⁻-binding rates with no substrate bound, by protonation state in shades of red (left), or with glutamate or aspartate (substrate) bound for the single and double protonation states (middle, right). During modification, unbinding constants were simultaneously altered to keep the ratio (i.e., the binding affinity) constant. Amplitudes are the optimized value plus 50 logarithmically distributed points between 1 and the ligand binding limit of $5 \times 10^9$; binding rates are at -160 mV and normalized to a [Cl⁻] of 40 mM. The same RSS derived from a wide range of rate values indicates that the $K_D$ determines the RSS, with no major role for individual binding/unbinding rate constants.
(TIF)

**S1 Table. Modeled WT VGLUT1$_{PM}$ Cl⁻ current rate constant, $z$, and $d$ amplitudes.** Binding steps and bound ions are shown in red for protons and green for Cl⁻. Unprotonated channel opening is limited to a rate constant of $1.00 \, s^{-1}$.
(XLSX)

**S2 Table. Modeled H120A VGLUT1$_{PM}$ Cl⁻ current rate constant, $z$, and $d$ amplitudes.** Binding steps and bound ions are shown in red for protons and green for Cl⁻. Unprotonated channel opening is limited to a rate constant of $1.00 \, s^{-1}$.
(XLSX)

**S3 Table. Rate constant, $z$, and $d$ amplitudes of the WT VGLUT1$_{PM}$ active transport model.** Binding steps and bound ions are shown in red for protons, green for Cl⁻, blue for glutamate, and yellow for aspartate. States that are inward-facing are labeled with $_i$ and outward-facing ones with $_o$. The doubly protonated transition between inward and outward transporter conformations is limited to $1.00 \, s^{-1}$ when glutamate is bound.
(XLSX)

 

## Acknowledgments

We thank Dr. Shigeo Takamori for kindly providing pcDNA3.1-rVGLUT1. The authors gratefully acknowledge the computing time on the supercomputer JURECA at Forschungszentrum Jülich granted through JARA under grant no. mpogt.

## Author contributions

**Conceptualization:** Bart Borghans, Christoph Fahlke.

**Data curation:** Bart Borghans, Daniel Kortzak.

**Formal analysis:** Bart Borghans, Daniel Kortzak, Piersilvio Longo, Jan-Philipp Machtens.

**Funding acquisition:** Christoph Fahlke.

**Investigation:** Bart Borghans, Daniel Kortzak, Bettina Kolen.

**Methodology:** Bart Borghans, Daniel Kortzak, Piersilvio Longo, Bettina Kolen, Jan-Philipp Machtens, Christoph Fahlke.

**Project administration:** Christoph Fahlke.

**Resources:** Christoph Fahlke.

**Software:** Bart Borghans, Daniel Kortzak, Piersilvio Longo, Jan-Philipp Machtens.

**Supervision:** Jan-Philipp Machtens, Christoph Fahlke.

**Validation:** Bart Borghans, Daniel Kortzak, Piersilvio Longo.

**Visualization:** Bart Borghans.

**Writing – original draft:** Bart Borghans, Christoph Fahlke.

**Writing – review & editing:** Christoph Fahlke.

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
