## [Decision Letter · Decision Letter 0]

3 Feb 2025

PCOMPBIOL-D-24-01761

Allosteric modulation of proton binding confers Cl- activation and glutamate selectivity to vesicular glutamate transporters

PLOS Computational Biology

Dear Dr. Fahlke, 

Thank you for submitting your manuscript to PLOS Computational Biology. After careful consideration, we feel that it has merit but does not fully meet PLOS Computational Biology's publication criteria as it currently stands. Therefore, we invite you to submit a revised version of the manuscript that addresses the points raised during the review process.

Please submit your revised manuscript within 30 days Apr 05 2025 11:59PM. If you will need more time than this to complete your revisions, please reply to this message or contact the journal office at ploscompbiol@plos.org. Please include the following items when submitting your revised manuscript:

We look forward to receiving your revised manuscript.

Kind regards,

Suhita Nadkarni, Ph.D

Academic Editor

PLOS Computational Biology

Lyle Graham

Section Editor

PLOS Computational Biology

**Additional Editor Comments :**

Both the reviewers commend the rigor and novelty of your study, particularly its use of a global fitting algorithm to resolve complex kinetic parameters and its mechanistic insights into chloride’s role in VGLUT function. The reviewers have suggested several points for improvement, like checking pH jump currents in non-transfected cells, clarifying units in figures, solution exchange time resolution, evidence for Cl- roles, explaining a low pKa value, and expanding on the kinetic modeling details. There's also a major concern about validating the model by simulating data with noise and refitting. One of the reviewers finds the conclusions plausible but raises concerns about the model's complexity and parameter determination. They also list minor issues like figure legends, fitting algorithm details, showing actual fits, and sharing genetic algorithm code. Detailed reviews are attached below.

**Journal Requirements:**

At this stage, the following Authors/Authors require contributions: Christoph Fahlke. Please ensure that the full contributions of each author are acknowledged in the "Add/Edit/Remove Authors" section of our submission form.

4) Please amend your detailed Financial Disclosure statement. This is published with the article. It must therefore be completed in full sentences and contain the exact wording you wish to be published.

5) Please ensure that the funders and grant numbers match between the Financial Disclosure field and the Funding Information tab in your submission form. Note that the funders must be provided in the same order in both places as well. Currently, the order of the grants is different in both places.

Please indicate by return email the full and correct funding information for your study and confirm the order in which funding contributions should appear. Please be sure to indicate whether the funders played any role in the study design, data collection and analysis, decision to publish, or preparation of the manuscript.

**Reviewers' comments:**

Reviewer's Responses to Questions

Reviewer #1: This is an interesting manuscript detailing the complex mechanism of VGLUT electrophysiological function. Due to the associated chloride current, transporter function can be analyzed, despite the actual glutamate-dependent transport current being too small for current recording. The authors propose a mechanism in which Cl- and protons bind to the transporter to form a large variety of states with differential Cl- conducting properties. As such, Cl- and protons allosterically affect each other’s binding properties. The important aspect here is that the complex kinetic schemes have been fit with a global fitting algorithm, which allows kinetic parameters to be determined for a large number of experimental current traces at the same time, with high confidence. This is a major advance in this study, since it takes away uncertainty when fitting individual current traces only. Overall, this work is very thorough and the quality of the data and analysis is high. However, I have some suggestions to further improve the manuscript:

1) Do pH jumps generate unspecific current in non-transfected cells? This data should be shown.

2) Fig. 1D-F: The y-axis current scale lacks the unit (my guess it is nA?).

3) What is the time resolution of the piezo-based solution exchange system?

4) The kinetic model in Fig. 2 predicts that Cl- both permeates the channel and modulates channel opening. What is the evidence for this?

5) Page 26: pKa of 0.3 is a very low value, indicating a strong acid. This is well outside of the pKa range of ionizable protein side chains. How can this be explained?

6) I don’t believe the kinetic modeling is described in sufficient detail. 1) The differential rate equations that (I suppose ) are integrated analytically at least need to be explained. 2) The genetic algorithm for the global fit is mentioned several times, but not explained in enough detail for this reviewer to understand what was actually done. Simple citing a machine learning paper as a reference is not sufficient.

Reviewer #2: The authors found that Cl- accelerates H+-glutamate exchange, mainly by making the glutamate binding site accessible to the cytoplasm, and by stimulating the inward translocation after substrate release to the vesicular lumen. While it was previously known that chloride ions play a role in vesicular glutamate transport, the specific mechanistic detail about Cl- making the glutamate binding site more accessible to the cytoplasm and stimulating inward translocation appears to be a novel finding. This provides a much more detailed molecular understanding of how Cl- influences the transport process.

In this manuscript, it was shown that Cl- increases the pKa of the second protonation site of the aspartate-bound inward-facing transporter, thereby promoting double protonation and preventing H+-coupling of aspartate transport. This finding about Cl- increasing the pKa of the second protonation site, specifically in the aspartate-bound inward-facing state, is novel. While the role of protonation in transport was known, this precise mechanistic detail about how Cl- affects the protonation state and prevents H+-coupling for aspartate specifically appears to be a novel discovery.

The authors of this manuscript showed that during the activation of VGLUT1 anion channels, Cl- stimulates the first protonation of the closed channel, accelerates the opening of the singly protonated closed channel, and then stabilizes the open state by virtually abolishing channel closure regardless of protonation. This detailed step-by-step mechanism of how Cl- affects channel activation through multiple stages appears to be novel. While VGLUT1's channel activity was known, this level of mechanistic detail about Cl-'s role in the process seems to be a new contribution.

The H120A point mutation increases the H+-binding affinity of the apo state well above the levels observed in the WT, especially in the absence of Cl-, and impairs Cl- binding by reducing its binding to the closed channel and increasing unbinding from the open channel for all protonation states. The specific findings about how this mutation affects H+-binding affinity and Cl--binding/unbinding kinetics in different channel states appear novel. This provides new structural insights into the role of this residue in the transport mechanism.

While I find the conclusions of the authors plausible I am not sure whether I can fully follow their approach. The authors use quite complex models and retrieve the parameters utilizing a genetic fitting algorithm. Given the complexity of these models (i.e more than 20 rate constants) and their many parameters (each rate constant is defined by the rate 0 mV, valence, and symmetry factor) I find it surprising that the values of the parameters are so well defined by the available experimental data.

Major concern:

One test that the authors should conduct to validate their approach is to simulate data with their model, add realistic noise, and evaluate if by fitting these data with genetic fitting alorithms the authors can get back the rate constants that they used for simulation.

Minor concerns:

1.) I would like to point out that the Butler-Volmer equation (ref. 40) is a specific case within the broader framework of transition state theory (TST) and the linear free energy relationship (LFER). TST and LFER provide a more general and fundamental theoretical foundation for understanding reaction rates and kinetic processes. The Butler-Volmer equation specifically deals with electrode kinetics and electrochemical reactions, while TST/LFER can describe virtually any type of chemical transition/reaction. Thus, I suggest to replace ref. 40 with the work of Boytsov et al. (DOI: 10.3389/fphys.2023.1166450).

2.) Additionally, it is necessary to expand the figure legends. Specifically, the description for the rectangles in Fig. 4 and the following figures. I got confused with the rectangular boxes. They may represent the variance of a parameter or they represent the minimum and the maximum of the parameter variants. I suggest plotting the parameter variants consistently (violin diagram)

3.) Please provide a better description of the fitting algorithm and the statistical analysis that pertains to the fit.

• What is meant by a variant of a parameter?

• Were all the data fit simultaneously or each experiment individually?

• What was the function used for minimization (sum of least squares?)?

• What exactly is the meaning of simulated metrics?

• Describe the weighting procedure in more detail.

4.)Fluxes through kinetic models can also be calculated using continuous-time Markov state modeling (Hill T.L. Free Energy Transduction and Biochemical Cycle Kinetics. Dover Publications; Mineola, NY, USA: 2004. ). Why is it that the authors decided to do this using discrete Markov state modeling? To this point: please provide a better description of this conversion (what is the lag time used etc.)

5.) Please provide a table with all rate constants of the two models (can be put into the supplement)

6) It would be helpful to indicate the positions which might be protonated (i.e., protonation relevant for transport)

7) I suggest showing the actual fit and not only the confidence interval (e.g., fig 1B, C, etc.).

8) The authors provide a Python code for the kinetic model, which is available publicly. However, I suggest providing a code for the genetic fit algorithm as well

**Have the authors made all data and (if applicable) computational code underlying the findings in their manuscript fully available?**

Reviewer #1: Yes

Reviewer #2: Yes

PLOS authors have the option to publish the peer review history of their article (what does this mean? ). If published, this will include your full peer review and any attached files.

**Do you want your identity to be public for this peer review?** For information about this choice, including consent withdrawal, please see our Privacy Policy .

Reviewer #1: No

Reviewer #2: No

**Figure resubmission:**
---

## [Decision Letter · Decision Letter 1]

9 Jun 2025

Dear Dr. Fahlke,

We are pleased to inform you that your manuscript 'Allosteric modulation of proton binding confers Cl- activation and glutamate selectivity to vesicular glutamate transporters' has been provisionally accepted for publication in PLOS Computational Biology.

Best regards,

Suhita Nadkarni, Ph.D

Academic Editor

PLOS Computational Biology

Lyle Graham

Section Editor

PLOS Computational Biology

Reviewer's Responses to Questions

**Comments to the Authors:**

Reviewer #1: This is a comprehensive revision of the original manuscript, answering all of my questions. I have no additional concerns.

Reviewer #2: The authors answered all my questions.

**Have the authors made all data and (if applicable) computational code underlying the findings in their manuscript fully available?**

Reviewer #1: Yes

Reviewer #2: Yes

PLOS authors have the option to publish the peer review history of their article (what does this mean? ). If published, this will include your full peer review and any attached files.

**Do you want your identity to be public for this peer review?** For information about this choice, including consent withdrawal, please see our Privacy Policy .

Reviewer #1: No

Reviewer #2: No

---

## [Editor Report · Acceptance letter]

PCOMPBIOL-D-24-01761R1

Allosteric modulation of proton binding confers Cl- activation and glutamate selectivity to vesicular glutamate transporters

Dear Dr Fahlke,

I am pleased to inform you that your manuscript has been formally accepted for publication in PLOS Computational Biology. Your manuscript is now with our production department and you will be notified of the publication date in due course.

With kind regards,

Olena Szabo
